# Feedback inhibition and its control in an insect olfactory circuit

**Subhasis Ray, Zane N Aldworth, Mark A Stopfer\***

Section on Sensory Coding and Neural Ensembles, NICHD, NIH, Bethesda, United States

**Abstract** Inhibitory neurons play critical roles in regulating and shaping olfactory responses in vertebrates and invertebrates. In insects, these roles are performed by relatively few neurons, which can be interrogated efficiently, revealing fundamental principles of olfactory coding. Here, with electrophysiological recordings from the locust and a large-scale biophysical model, we analyzed the properties and functions of GGN, a unique giant GABAergic neuron that plays a central role in structuring olfactory codes in the locust mushroom body. Our simulations suggest that depolarizing GGN at its input branch can globally inhibit KCs several hundred microns away. Our in vivorecordings show that GGN responds to odors with complex temporal patterns of depolarization and hyperpolarization that can vary with odors and across animals, leading our model to predict the existence of a yet-undiscovered olfactory pathway. Our analysis reveals basic new features of GGN and the olfactory network surrounding it.

## Introduction

Olfactory information is transformed dramatically as it travels from the periphery to higher order brain centers. Multiple types of olfactory receptor neurons can respond to a given odor with vigorous bursts of action potentials, while neurons deeper in the brain, in the pyriform cortex (vertebrates) or mushroom body (insects), respond to the same odor with only a spike or two (*Laurent and Naraghi, 1994b*; *Friedrich and Laurent, 2001*; *Perez-Orive et al., 2002*; *Cang and Isaacson, 2003*; *Bathellier et al., 2008*; *Poo and Isaacson, 2009*). In these higher order neurons, information about odors is represented sparsely by the identities of the active neurons (population coding) and in the timing of the few spikes elicited in those neurons (temporal coding) (*Perez-Orive et al., 2002*; *Poo and Isaacson, 2009*; *Stettler and Axel, 2009*; *Gupta and Stopfer, 2014*). In both vertebrates and invertebrates multiple mechanisms interact to mediate these transformations, including important inhibitory contributions from GABAergic neurons (*Luna and Pettit, 2010*; *Papadopoulou et al., 2011*; *Lin et al., 2014*; *Palmer and Harvey, 2014*; *Large et al., 2016*; *Large et al., 2018*). Here, with intracellular recordings and a new large-scale biophysical model that includes tens of thousands of cells and spans multiple layers of olfactory processing, we focus on a singularly important inhibitory neuron to investigate the roles of input activity and feedback inhibition in creating a sparse spatio-temporal odor representation in a higher order brain center. Together, our recordings and models point to new functions, neural connectivity patterns, and mechanisms that underlie transformations in the format of olfactory information.

The locust olfactory system is well-studied because it is accessible, robust, and relatively simple. In the absence of environmental odors, olfactory receptor neurons (ORNs) in the antenna are spontaneously active. Odorants elicit increases or decreases in the spontaneous firing rates of ORNs, and simple patterns of spikes with sequences of excitation and inhibition, distributed heterogeneously across the ORN population. These responses vary with the odor and its concentration. Spontaneous activity in ORNs drives background spiking in PNs (*Joseph et al., 2012*). PNs respond to the variety of odor-evoked responses in ORNs with sometimes-elaborate firing patterns that are further shaped

**\*For correspondence:**
stopferm@mail.nih.gov

**Competing interests:** The authors declare that no competing interests exist.

by inhibition from the antennal lobe's local interneurons (LNs) (*Raman et al., 2010*). Spikes in PNs are also coaxed into rhythmic waves by fast reciprocal interactions between excitatory PNs and inhibitory LNs (*MacLeod and Laurent, 1996*; *Bazhenov et al., 2001*). Odor-elicited firing patterns distributed across the population of PNs are informative about the identity, concentration, and timing of the odor (*Laurent et al., 1996*; *Stopfer et al., 2003*; *Brown et al., 2005*). This information is carried by PNs to the mushroom body and the lateral horn.

The primary neurons within the mushroom body are Kenyon cells (KCs). Unlike the volubly spiking PNs, KCs are nearly silent at rest and respond very selectively to odors by generating very few spikes (*Laurent and Naraghi, 1994b*; *Perez-Orive et al., 2002*; *Joseph et al., 2012*). Thus, any given odor evokes responses in a small fraction of the KC population, and any KC responds to a small set of odors (*Perez-Orive et al., 2002*; *Stopfer et al., 2003*). This sparseness of activity in KCs is thought to arise mainly from two factors: specialized membrane conductances that amplify only strong, synchronized input; and a feedback circuit that tamps down their spiking with cyclic inhibition (*Perez-Orive et al., 2002*; *Demmer and Kloppenburg, 2009*; *Papadopoulou et al., 2011*; *Lin et al., 2014*). In the locust the main sources of this inhibition are the giant GABAergic neurons (GGNs), one on each side of the brain (*Leitch and Laurent, 1996*; *Papadopoulou et al., 2011*; *Gupta and Stopfer, 2012*). Thus, GGN plays a central role in shaping neural codes for odors.

GGN spans much of each brain hemisphere and branches very widely (*Figure 1a*). It is reported to receive excitatory input from all 50,000 KCs at synapses within the mushroom body's α lobe and, in turn, provide inhibitory feedback to all KCs 400–500 µm away within the calyx. In addition, GGN receives inhibitory input from a spiking neuron aptly named 'Inhibitor of GGN' (IG) which itself receives inhibition from GGN (*Figure 1a*, right) (*Papadopoulou et al., 2011*). GGN is a non-spiking interneuron. Odor presentations, spiking in KCs, and intracellular current injections have all been shown to depolarize GGN, but none of these stimuli causes GGN to generate spikes; even strong intracellular current injections into GGN elicit only passive responses (*Leitch and Laurent, 1996*; *Papadopoulou et al., 2011*); and our own observations).

GGN's structure likely shapes its function. GGN is very large, and along its path from the α lobe to the calyx, its initially thick processes divide at myriad branch points into vanishingly thin fibers. Cable theory applied to neurons (*Rall, 1964*) predicts that a passive voltage signal within such a structure will attenuate dramatically as it encounters cytosolic resistance along the neurites, will attenuate further as it divides at the neuronal arbor's branch points, and will leak out through ionic channels in the cell membrane. Thus, it seemed possible that GGN might lack the biophysical capacity to transmit signals long distances throughout its expansive arbors. It has been shown that GGN inhibits KCs (*Papadopoulou et al., 2011*). However, if signals originating in GGN's α lobe attenuate to the extent that they cannot (as proposed) effectively and globally hyperpolarize KCs via synapses 400–500 µm away in the calyx, then GGN must be inhibiting KCs through a different mechanism; perhaps, for example, through local interactions entirely within the mushroom body calyx.

The extent of signal attenuation throughout GGN has not been measured directly. Our own efforts to achieve this by making simultaneous recordings from multiple branches of GGN, and by using optical techniques, yielded results that were difficult to interpret. Therefore, we developed a realistic computational model of GGN to characterize signal attenuation throughout its structure. Prior studies in invertebrates have shown that 2–5 mV depolarizations in nonspiking interneurons suffice to evoke changes in the membrane potentials of their post-synaptic neurons (*Burrows and Siegler, 1978*; *Manor et al., 1997*). Our model showed that, although electrical signals do undergo substantial attenuation as they travel through GGN, signals arising in its α lobe branches appear to remain strong enough to provide global inhibition to KCs in the calyx.

Because GGN plays an outsized and central role in the olfactory system, investigating its responses to odors can illuminate broader issues of olfactory function. Thus, to further understand the network determinants of GGN's responses to odors, we recorded from it in vivo while delivering a variety of odors to the animal, and then developed a large-scale model including KCs, IG, and realistic olfactory inputs from PNs, to investigate the types of network activity needed to generate the membrane potential patterns we observed in GGN. We identified two novel features in the olfactory network. First, to generate the types of membrane potential patterns we observed in GGN, the synaptic connection strengths onto KCs must be heterogeneous. Second, our in vivo recordings of GGN revealed novel, complex response patterns not previously documented, including periods of hyperpolarization, that vary with the odorant. Although GGN receives reciprocal feedback from

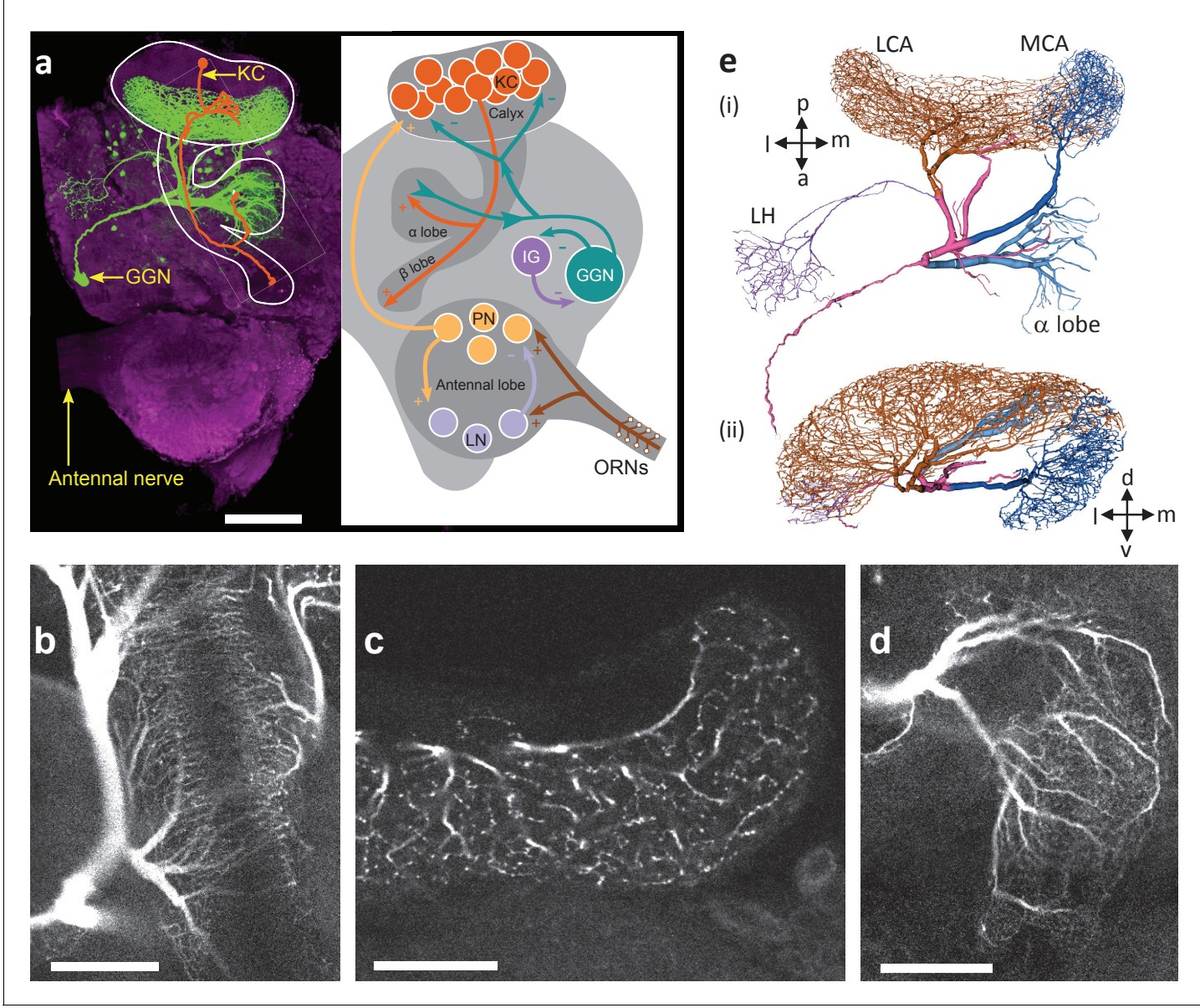

**Figure 1.** GGN is a very large neuron, one per hemisphere, spanning a large portion of the locust brain. (a) Left hemisphere: Dye filled GGN (green) in a locust brain (magenta, visualized with nc82 antibody) with an overlaid KC tracing (orange, filled in a different brain). Dorsal: towards the viewer; ventral: into the page). Right hemisphere: Circuit diagram of the locust olfactory system. Arrows show known synaptic connections, + excitatory, - inhibitory. Scale bar: 200 μm. (b) Very thin, feather-like neurites from GGN wrap around and penetrate the peduncle of the mushroom body; these neurites are not included in our reconstruction. (c) GGN's neurites in the calyx have many bouton-like protrusions (d) whereas GGN's α lobe branches are relatively smooth. (b–d) scalebars: 100 μm. (e) 3D reconstruction of the same neuron shown in panel (a) viewed from (i) ventral and (ii) posterior side of the brain. Major branches shown in different colors. LCA: lateral calyx, MCA: medial calyx, LH: lateral horn, a: anterior, p: posterior, d: dorsal, v: ventral, l: lateral and m: medial.

The online version of this article includes the following video for figure 1:

**Figure 1—video 1.** 3D view of the confocal stack of the dye filled GGN in *Figure 1a*.
https://elifesciences.org/articles/53281#fig1video1

IG (*Papadopoulou et al., 2011*), the periods of hyperpolarization could not be explained by disinhibition of IG from GGN. Instead, our model predicts that this behavior could arise if, in addition to receiving input from GGN, IG also receives direct excitation from another, presently unknown odor-activated pathway. Additionally, our model replicated emergent features of the olfactory network

not explicitly programmed into it, such as the appearance of a small portion of KCs spiking at relatively high rates.

Together, the results of our in vivo recordings and large-scale realistic computational models provide a more complete understanding of how different parts of the olfactory system interact. To generate odor-specific temporally patterned responses in GGN and in the mushroom body, temporally-patterned odor evoked excitation from PNs, feedback inhibition from GGN, and inhibition of GGN by odor-driven IG must all cooperate.

## Results

### GGN morphology

A valuable use of computational modeling is to answer questions about biological systems when those systems are too large, too complex, or too difficult to study with the tools of physiology alone. Earlier computational studies of the insect olfactory system used relatively simple models of neurons such as integrate-and-fire or map-based approaches that collapse entire neuronal structures into a single point (*Perez-Orive et al., 2004*; *Papadopoulou et al., 2011*; *Arena et al., 2015*; *Kee et al., 2015*; *Peng and Chittka, 2017*). However, GGN's giant size, elaborate branching, and passive membrane properties raised questions about its function that could only be addressed by considering properties determined by its morphology. Thus, to understand how the size and shape of GGN affects electrical signal propagation, we constructed a detailed morphological model (available at neuromorpho.org).

To reconstruct the morphology of GGN, we first made intracellular recordings from it in vivo, filled it with dye, and obtained 3D confocal images of the dye-filled cell (*Figure 1a*, left; *Figure 1—video 1*). As previously shown, GGN has a reliable and unique location and morphology (*Weiss, 1978*; *Leitch and Laurent, 1996*; *Papadopoulou et al., 2011*; *Gupta and Stopfer, 2012*). On each side of the brain, its soma is located ventrally, just anterior to the optic nerve. A single neurite emerges from GGN's soma, travels toward the posterior and dorsal side of the brain, and splits there into two branches, one innervating the α lobe and the other the mushroom body. Extending outward, these branching neurites expand in width, becoming much thicker than the primary neurite. The mushroom body branch further divides into two thick processes that innervate the medial and the lateral calyx. A thin neurite emerging from the lateral calyceal branch loops back to the lateral horn, close to the soma, and splits there into many branches. We further observed, for the first time, myriad thin fibers that emerge from the stems of the calyceal branches, splitting into very fine feather-like neurites that wrap densely around the peduncle, with some investing the peduncle core (*Figure 1b*). The neurites in the calyx and lateral horn are dotted with many irregular bouton-like structures (*Figure 1c*) whereas the branches in α lobe are relatively smooth (*Figure 1d*).

In two animals we traced and reconstructed the morphology of GGN from confocal image stacks (*Figure 1e*). Analyzing these traces, we found that the maximum path length of the neuronal trees (i.e., the maximum distance between any two points on the neuronal tree when traversed along the neurites) was on the order of 2 mm, and the maximum physical length (i.e., Euclidean distance between any two points on the neuron in three-dimensions) was on the order of 1 mm. Some neurites at their thickest were nearly 20 μm in diameter. The total traced branch length (i.e., the sum of the lengths of all the neurites) was about 65 mm (although many vanishingly thin branches were too fine to trace). Compared to 96,831 vertebrate and invertebrate neurons cataloged in the neuromorpho.org database, GGN fell into the 99.75th percentile for total branch length, and the 99.95th percentile for number of branch points. It is a really big neuron.

### Signals traveling through GGN's arbors are predicted to attenuate significantly

To investigate GGN's electrical properties we constructed passive electrical model cells by transferring the morphology tracings of the two GGNs into the NEURON simulator (*Carnevale and Hines, 2006*). Simulations with both models produced qualitatively similar results; the model we describe here was derived from the second neuron we traced because it was imaged and mapped at higher resolution. To account for branches and changes in the diameters of processes that affect electrotonic distance, we segmented the model GGN into 5283 isopotential compartments. We set the

membrane resistivity in the model to 33 kΩ-cm$^2$ following published descriptions obtained from other non-spiking neurons in the locust (*Laurent, 1991*), the specific membrane capacitance to 1 μF/cm$^2$, the approximate value for cell membranes (*Curtis and Cole, 1938*; *Hodgkin and Huxley, 1952*; *Gentet et al., 2000*), and the cytoplasmic resistivity to 100 Ω-cm, a typical order-of-magnitude value for neurons (*Hodgkin and Rushton, 1946*; *Stuart and Spruston, 1998*; *Roth and Häusser, 2001*).

Feedback signals are thought to travel passively from GGN's α lobe branch to its calyceal branches. To test the extent of passive signal attenuation through GGN's structure, we simulated a voltage clamp experiment in the GGN model (*Figure 2a*). In vivo, intracellular recordings (*Papadopoulou et al., 2011*; and our own) show GGN's membrane potential rests at about −51 mV. Because strong odor stimuli depolarize GGN by about 10 mV in recordings made near the base of the α lobe branch (*Papadopoulou et al., 2011*) (also see Figure 4) we first stepped the clamp voltage in the α lobe branch to −40 mV, and, after holding it there for 450 ms, measured the resulting depolarizations throughout the model GGN (*Figure 2a* inset). Notably, the extent of signal attenuation was substantial and varied throughout the calyx, yielding depolarizations ranging from ~5–9 mV. The signal decreased with branch distance from the α lobe, leaving the least amount of signal at the medial portion of the lateral calyceal branch (*Figure 2b,c*, *Figure 2—video 1*).

Since excitatory input typically arrives in GGN not at one location, but rather at many synapses from many KCs, we sought to test a more realistic form of simulated input to the α lobe arbor of GGN. Thus, we provided nonhomogeneous Poisson spike trains through 500 excitatory model synapses; each synapse had a maximum rate of 20 spikes/s that ramped down linearly to 0 over a 500 ms interval (*Figure 2d,e*). This stimulus set was calibrated to generate a peak depolarization in the thick branches of GGN in the same range we observed in vivo. This more realistic test also revealed significant attenuation of voltage in the neuron's distant branches (*Figure 2e,f*) as well as delays in signal propagation (*Figure 2—figure supplement 1*).

Neither the membrane resistivity (RM) nor cytoplasmic (axial) resistivity (RA) has been measured definitively in GGN; yet, for a given morphology, these two parameters help determine signal attenuation. Thus, we explored a range of values for these two parameters with the voltage clamp simulation approach shown in *Figure 2a*. We based the RM value range on data obtained from many types of neurons provided by the neuroelectro.org database. For RA, neurophysiological data are sparse, so we explored broadly around the range of published values (*Hodgkin and Rushton, 1946*; *Stuart and Spruston, 1998*; *Roth and Häusser, 2001*). As expected, higher RA yielded greater signal attenuation, whereas higher RM yielded less signal attenuation (*Figure 2g*). This analysis suggests that signal transmission in GGN is robust: except for the most extreme values of this parameter range, signals from the α lobe remained strong enough throughout the calyx (>2–5 mV) to support synaptic transmission. The extent of depolarization throughout GGN's calyceal arbor varied with location, as quantified in the extended lower lobe in the violin plots in *Figure 2c,f and g*.

Branches of GGN receiving weaker signals from the α lobe would be expected to provide less inhibition to their postsynaptic KCs. In a simplified model in which all KCs were strongly stimulated by identical input from PNs, the amount of KC spiking was indeed negatively correlated with local GGN voltage deflections (*Figure 2—figure supplement 2*). However, in a more realistic model of the mushroom body network that included variable excitatory input from PNs and variable strengths of inhibitory synapses between GGN and KCs, we found the negative correlation between depolarizations measured at presynaptic locations throughout GGN and postsynaptic KC activity was small and likely negligible (*Figure 2—figure supplement 3*). This suggests GGN's inhibitory output has a surprisingly uniform influence across the population of KCs regardless of location.

## Feedback inhibition is predicted to expand the dynamic range of KCs

We used our model to test other basic properties of GGN and the olfactory network surrounding it. In vivo, feedback inhibition from GGN sparsens the odor-elicited responses of KCs by increasing the KC spiking threshold and by restricting KC spiking to brief temporal windows defined by the oscillatory cycle established in the AL (*Papadopoulou et al., 2011*; *Gupta and Stopfer, 2014*). Large-scale feedforward inhibition has previously been shown to expand the dynamic range of cortical neurons (*Pouille et al., 2009*), and in *Drosophila*, feedback inhibition from APL, the analog of GGN, expands the dynamic range of KCs (*Inada et al., 2017*). Whether feedback inhibition from GGN has a similar effect on KCs is unknown. To test this, we extended our model to include, for simplicity, a

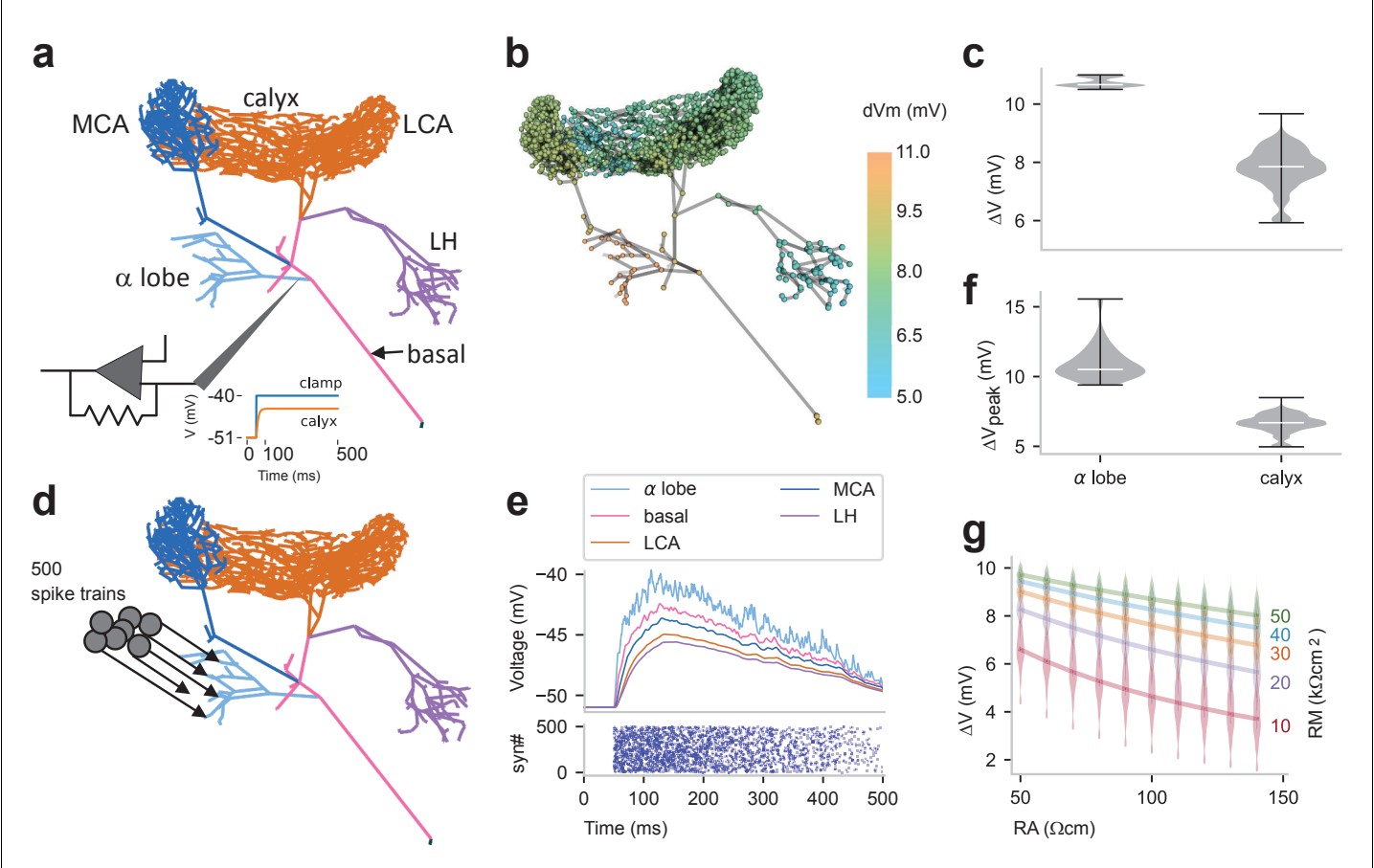

**Figure 2.** Passive voltage spread through GGN. (a) Simulation schematic: GGN's α lobe branch was clamped at its stem to −40 mV (from a holding voltage of −51 mV). (b) Conditions described in panel a lead to different steady state depolarizations of the GGN model at different locations; color indicates voltage change from resting potential; axial resistivity RA = 100 Ωcm; membrane resistivity RM = 33 kΩcm2. (c) Violin plots showing distribution of depolarization in the α lobe and the calyceal dendrites of GGN for conditions described in panel a; width of gray area indicates density, white line median, and whiskers data range. (d) Schematic of simulation of multiple, random synaptic inputs to GGN: 500 synapses were connected to GGN's α lobe branch. (e) Bottom: raster plot of incoming spike times at all 500 synapses. The spikes arrived at each synapse at random times at linearly decreasing rate, starting with 20/s down to 0 after 500 ms. Top: for conditions shown in panel d, membrane potentials at randomly sampled terminal neurites in different regions of GGN. (f) Violin plots of peak depolarizations in the α lobe dendrites and the calyceal dendrites in model described in panel d. (g) Violin plots of distribution of depolarizations in the calyx for different values of axial resistivity (RA, bottom) and membrane resistivity (RM, right) for the voltage clamp simulation described in panel a. Lines connect the median results for specified RMs (right).

The online version of this article includes the following video and figure supplement(s) for figure 2:

**Figure supplement 1.** Signal propagation time throughout GGN.

**Figure supplement 2.** KC spiking vs GGN voltage deflection in simulations where all KCs received identical PN input for different values of maximum synaptic conductances onto KCs.

**Figure supplement 3.** Correlations between number of spikes in a KC and various parameters influencing spiking: $\Delta V_m$ – peak voltage deflection of its presynaptic GGN segment, $\bar{g}_{GGN}$ – maximum conductance of the synapse from GGN, $\sum_{presynaptic} spikes_{PN}$ – total number of spikes over all its presynaptic PNs, $\sum_{presynaptic} \bar{g}_{PN}$ – sum of maximum conductance of all PN synapses, $\sum_{presynaptic} \bar{g}_{PN} \times spikes_{PN}$ – sum of the numbers of spikes in its presynaptic PNs weighted by the maximum conductances of their synapses onto this KC.

**Figure 2—video 1.** 3D view of depolarizations (color coded spheres) at various locations in GGN arbor in simulation described in *Figure 2a*.
https://elifesciences.org/articles/53281#fig2video1

single KC receiving feedback inhibition from GGN (*Figure 3a*). To simulate the KC in this test we used a single compartmental model with Hodgkin-Huxley type ion channels (*Wüstenberg et al., 2004*). Since a single KC would have negligible effect on GGN, we applied its spiking output to GGN's α lobe branch via 50,000 synapses. To avoid unrealistic, strong synchronous input to GGN,

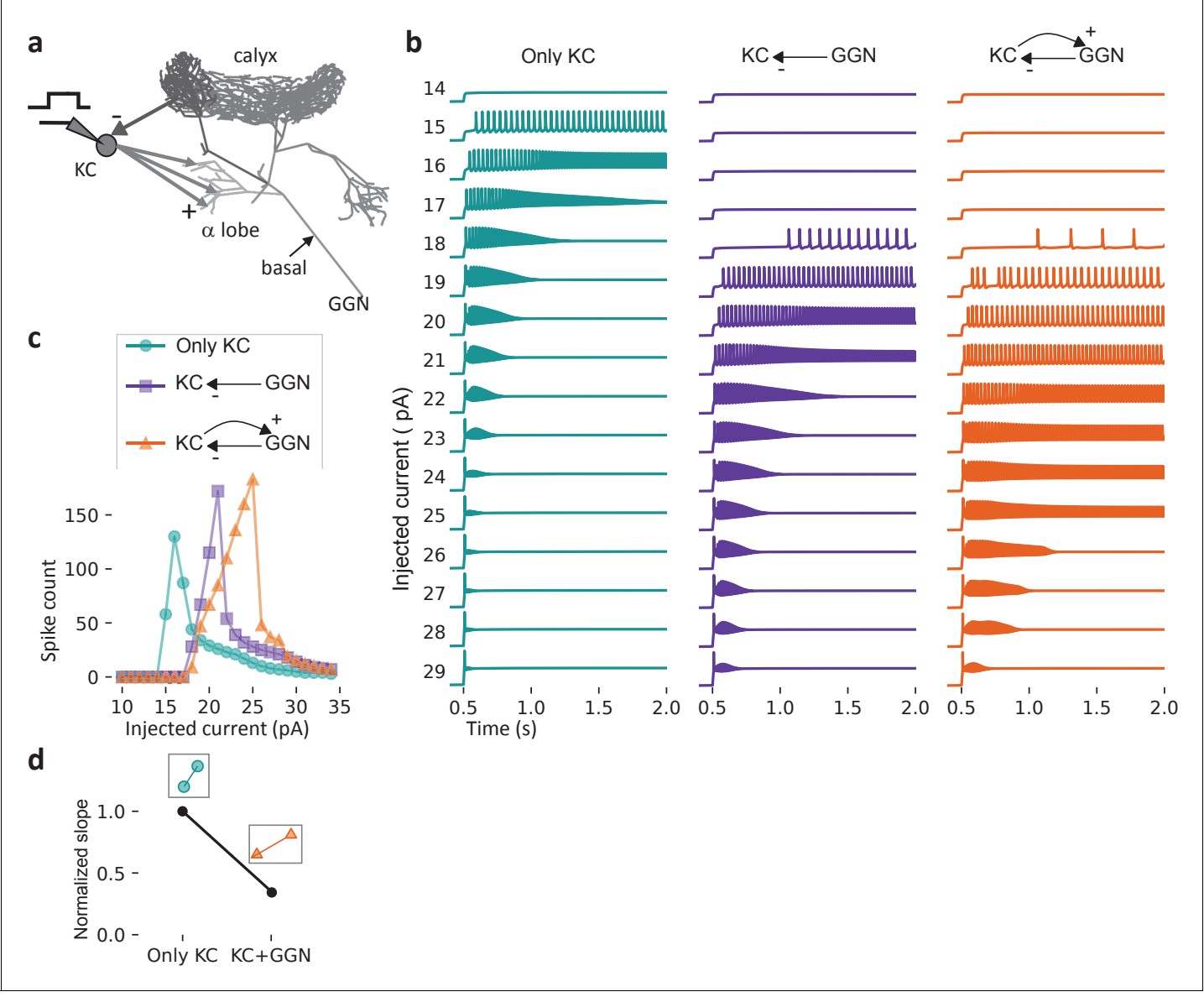

**Figure 3.** Feedback inhibition from GGN extends the dynamic range of KCs. (a) Model schematic: A single KC sends 50,000 excitatory synapses with random delays of 0–60 ms into GGN's α lobe branch. The KC receives feedback inhibition via a graded synapse from GGN's calyceal branch. A step current is injected into the KC from 0.5 to 3 s (up to 2 s is shown). (b) The model KC was tested with current pulses of increasing amplitude in three configurations: Isolated KC (left); KC receiving only spontaneous inhibition from GGN (middle), and KC receiving spontaneous and odor-elicited feedback inhibition (right). As the amplitude of the current pulse increases, the KC's spiking first increases, and then, as the membrane potential nears saturation, decreases. (c) Comparison of the number of spikes evoked by current steps in KC in the three configurations shown in b). (d) Slopes of the rising parts of the KC-only (left, cyan) and KC with GGN feedback (right, orange) curves in c), normalized by the KC-only slope.

The online version of this article includes the following figure supplement(s) for figure 3:

**Figure supplement 1.** Model KC responses to current step injections while inhibited by pseudo-dynamic clamp of GGN voltage from a network model show inhibition expands dynamic range of KCs.

we jittered the incoming spike times by applying random synaptic delays between 0 and 60 ms. Thus, after each spike generated by the model KC, GGN received 50,000 EPSPs spread over a 60 ms time window. We drove the KC model with a range of tonic current injections and compared its responses to those of an isolated KC model receiving the same input without feedback inhibition. As expected, baseline inhibition from spontaneous activity in GGN increased the KC's threshold for

spiking. Notably, though, the GGN-coupled KC continued to spike over a much larger range of current injection than the isolated KC, which quickly saturated to a level where it could no longer spike (*Figure 3b,c*). A similar result was obtained when we tested the KC by applying simulated GGN inhibition from a model of the olfactory network described later (*Figure 3—figure supplement 1*). These results suggested that feedback inhibition from GGN allows an individual KC to function effectively over a larger dynamic range of inputs. To quantify our results, we used a standard analysis from control systems literature in which dynamic range is characterized by the slope of the input-response curve, which quantifies the effectiveness of input for eliciting output. Expanding the dynamic range makes the slope of the input-response curve shallower, as we observed in our model (*Figure 3c and d*; see Materials and methods for the slope calculation). Thus, our model predicts that feedback inhibition from GGN expands the dynamic range of KCs.

## GGN responses can be complex, including hyperpolarization

Our recordings made in vivo from GGN frequently revealed depolarizations throughout an odor presentation, often with stronger depolarizing peaks upon the odor's onset and offset (*Figure 4*, Animal 1, hexanol) (see also *Papadopoulou et al., 2011*). Notably, though, our recordings from GGN sometimes revealed more complex odor-elicited responses than previously reported, including combinations of depolarization and hyperpolarization (*Figure 4*, Animal 1, hexanal). Moreover, we found that the same GGN could respond differently when different odors were presented; for example, GGNs from Animals 2 and 3 shown in *Figure 4* depolarized in response to one odor and hyperpolarized in response to another. Also, GGNs in different animals could respond differently to the same odor (*Figure 4*, hexanal). Almost a quarter of the odor-GGN pairs in our in vivo recordings showed reliable hyperpolarizations at some point in the odor response (40 out of 169). Earlier computational models (*Papadopoulou et al., 2011*; *Kee et al., 2015*) did not attempt to reproduce the variety of GGN responses we observed in vivo, including hyperpolarizations. To better understand the mechanisms underlying GGN's odor-elicited responses (and by extension, novel features of olfactory

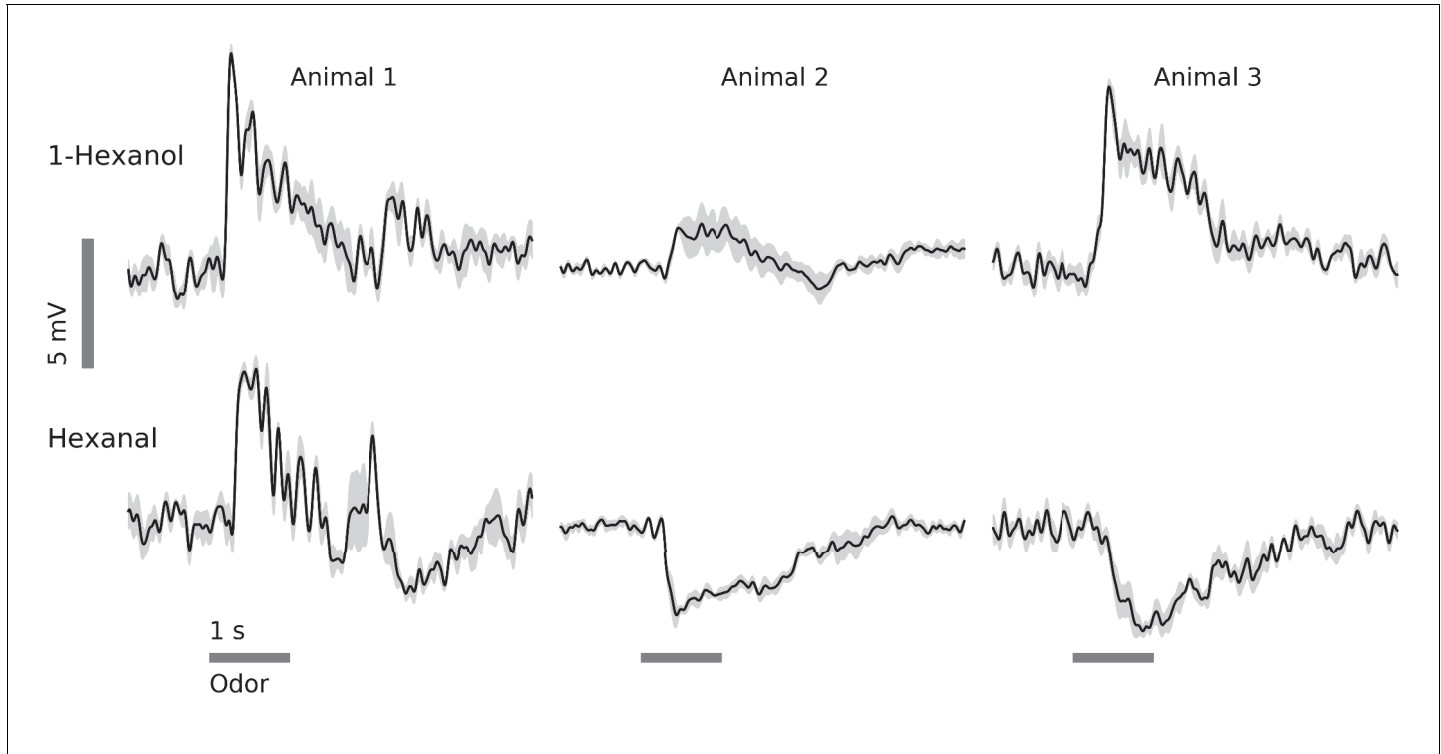

**Figure 4.** In vivo, GGN's responses to odors vary with odor and animal. Examples from 3 animals and two odors (horizontal gray bars, 1 s) (black traces: average of 5 trials, gray: standard error of the mean, data low-pass filtered at 49 Hz). Response features include depolarization or hyperpolarization upon stimulus onset and/or offset, and depolarization or hyperpolarization evoked by different odors in the same GGN.

circuitry), we placed our GGN model at the center of a more extensive mushroom body olfactory network.

## Sustained GGN responses require heterogeneous input to KCs

We sought to identify olfactory network characteristics required to generate realistic responses in GGN. Because GGN plays a central role in the olfactory system, this analysis could reveal gaps in our understanding of the network. We first tested whether simply simulating the average statistics of olfactory input to the mushroom body network would elicit responses from GGN like those observed in vivo. To test this, we extended our detailed GGN model with a full population of 50,000 simulated KCs. We used a relatively simple NEURON version of a single compartmental KC model (*Wüstenberg et al., 2004*). This approach was sufficient because KCs are very small, have few dendritic branches, and generate action potentials – in contrast to the large, complex, and passive GGN, the morphologies of individual KCs are unlikely to differentially influence their odor coding properties. Each model KC was connected to GGN in the α lobe via an excitatory synapse, and, in turn each KC received inhibitory input from a random segment of GGN in the calyx via a graded synapse (*Figure 5a*). To provide excitatory input to the KCs, the firing patterns of 830 PNs were simulated as spike-trains, and, following observations made in vivo (*Jortner et al., 2007*), each KC was stimulated by 50% of these spike trains via excitatory synapses. The spike trains were designed to follow the statistics of PNs recorded in vivo (*Mazor and Laurent, 2005*; *Jortner et al., 2007*). Thus, 77% of the PN spike trains were assigned a spontaneous firing rate of 2.6 spikes/s; during odor stimulation, 20% of these PNs were switched to 20 spikes/s modulated by the 20 Hz oscillations generated in the antennal lobe and reflected in the local field potential (LFP), and 10% were inhibited (no spikes) (*Figure 5b*). In this model all the synapses between two given cell types had the same conductance. Simulations of this model produced a few highly synchronized bouts of spiking in the KC population (*Figure 5c*), and corresponding isolated peaks in GGN's membrane potential (*Figure 5d*). These unrealistic responses were similar to those generated by earlier models (*Papadopoulou et al., 2011*; *Kee et al., 2015*).

Since this simplified model network did not reproduce the long-lasting depolarization of GGN we had observed in vivo, we introduced more biological detail. Synaptic strengths follow a lognormal distribution (*Song et al., 2005*; *Loewenstein et al., 2011*; *Buzsáki and Mizuseki, 2014*). After we adjusted our network model to include this property, some KCs became weakly inhibited, allowing them to fire more volubly. We also adjusted the input to KCs provided by PNs. In our simplified model, input to the KCs included spikes from a fixed set of PNs that were all active throughout the duration of the odor stimulus. However, in vivo, spiking patterns including periods of excitation and inhibition in PNs evolve over the course of an odor presentation, and different PNs respond to the same odor in different ways (*Laurent and Davidowitz, 1994a*; *Stopfer et al., 2003*; *Mazor and Laurent, 2005*); thus, a given odor presentation activates sets of KCs that shift in composition over the course of the response.

To simulate these diverse responses, we divided the model's PN population into five groups: four groups that responded to the stimulus and one that did not. In each of the responsive groups, odors activated changing subsets of PNs, updated with each LFP cycle (*Figure 5e*). Lacking the heterogeneity in synaptic strengths onto KCs described above, even this complex and changing input activity pattern elicited unrealistically synchronized bouts of activity in KCs, resulting in unrealistic isolated peaks in GGN's simulated membrane potential (*Figure 5f,g*). Increasing the strength of input from PNs allowed KCs to spike throughout the stimulus duration and generate sustained depolarization of GGN, but, under this condition, the number of spiking KCs became much higher than observed in vivo (*Perez-Orive et al., 2002*). Increasing the diversity of synaptic delays from PNs to KCs did not alleviate overly-synchronized spiking in the KC population if heterogeneity in synaptic strengths and temporal patterns of PN activity were omitted from the model (*Figure 5—figure supplement 1*). However, a model combining heterogeneous connectivity with structured PN firing patterns gave rise to GGN voltage traces that included sustained depolarization and temporal dynamics more characteristic of responses we had observed in vivo (*Figure 5h,i*). Notably, even the steady PN activity shown in *Figure 5b* could produce sustained depolarization in GGN when the synaptic strengths were diversified (*Figure 5—figure supplement 2*). This analysis, centered on GGN's responses, revealed important constraints for modeling the olfactory system.

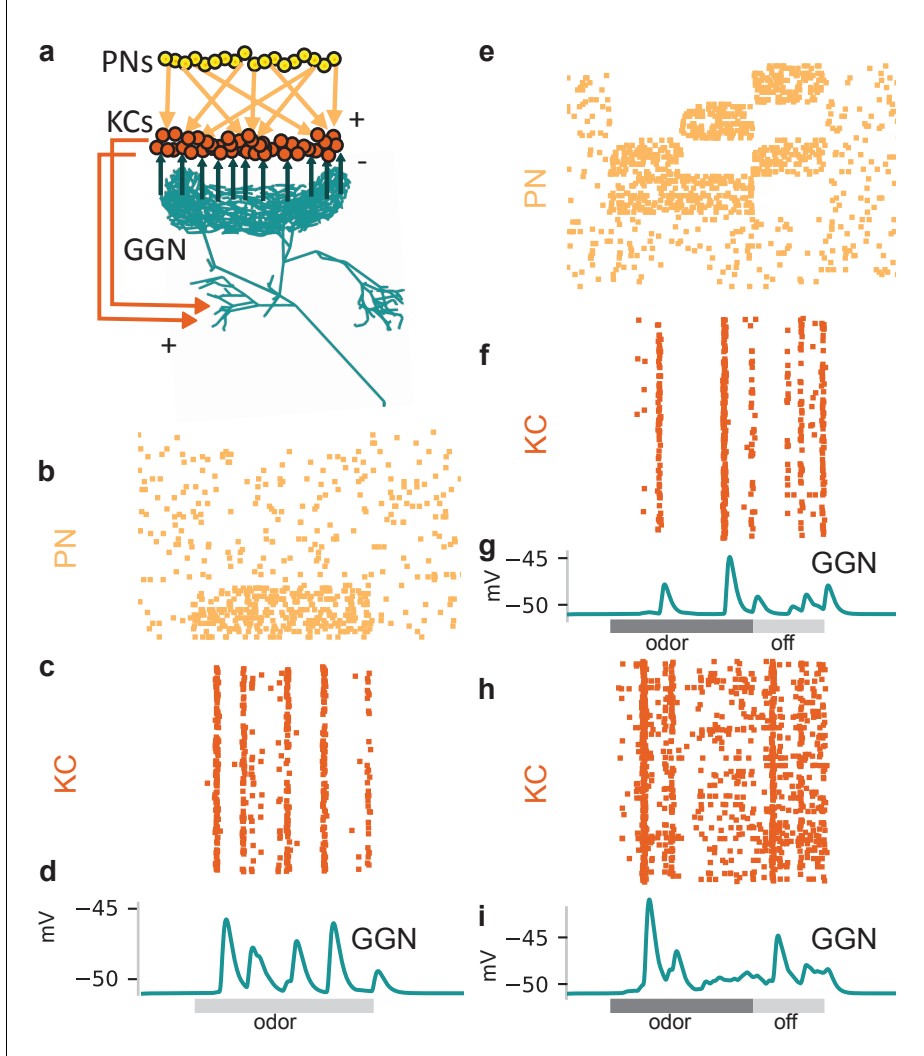

**Figure 5.** Tuning the olfactory network by reference to GGN's olfactory responses required heterogeneous synaptic strengths onto KCs and structured patterns of activity in PNs. (**a**) Schematic of mushroom body network model. Each of the 50,000 KCs receives input from 50% of the 830 PNs, which are modeled as spike trains. All KCs excite GGN in its α lobe branch and receive inhibition from a random calyceal branch of GGN. (**b–d**) Model with simplified, homogenous firing patterns in PNs and uniform synaptic strengths generates unrealistic membrane potential in GGN. (**b**) Raster plot of model PN spike trains (67 shown); dots in each row mark spike times in a PN. (**c**) Raster plot of spike trains evoked in the KCs (397 shown) when all of them receive identically strong inhibitory connections from GGN. (**d**) Unrealistic membrane potential in GGN features a few peaks corresponding to highly synchronized bouts of activity in KCs. Light gray bar: 1 s odor stimulation. (**e–g**) Model in which subpopulations of PNs have different temporal patterns of spiking during and after odor stimulation generates unrealistic membrane potential in GGN. (**e**) Rasters show different firing patterns in different PNs. (**f**) Simulation of model with PN activity pattern in panel e) along with uniform synaptic strengths onto KCs generate synchronized spiking in KCs and (**g**) unrealistic membrane potential in GGN. Simulation of a model with both structured PN activity patterns in panel e) and heterogeneous synaptic strengths gives rise to (**h**) temporally diffuse spiking in KC population and (**i**) sustained depolarization of GGN similar to that observed in vivo (e.g. *Figure 4* animal 1). Dark gray bar: 1 s odor stimulation; light gray bar: 0.5 s 'off response' period.

The online version of this article includes the following figure supplement(s) for figure 5:

**Figure supplement 1.** Introducing diverse delays in PN to KC synapses does not alleviate the unrealistic extent of synchronization of KC firing.

**Figure supplement 2.** Model with steady activity in a fixed set of PNs (as in *Figure 5b*) can produce sustained depolarization of GGN when the synaptic strengths are lognormally distributed.

## Some KCs fire at high rates in a GGN-centered olfactory network model

The distribution of firing rates of the KC population in our simulations showed that, while most KCs produced 0–2 odor-elicited spikes, a few KCs spiked much more (*Figure 6a* and *Figure 6—figure supplement 1*). To compare the firing rate distribution of KCs in the model to that in vivo, we made patch clamp recordings from 147 KCs in 114 animals, obtaining results from 707 KC-odor pairs. On average, the spontaneous firing rates of these real KCs were very low (~0.09 Hz) and reached only somewhat higher rates during odor presentation and immediately after odor termination (~0.15 and~0.16 Hz, respectively, *Figure 6b*), as previously observed (*Perez-Orive et al., 2002*; *Gupta and Stopfer, 2014*). Some odor-elicited responses in KCs, though, consisted of many more spikes. *Figure 6c* shows a representative example of a hyperactive response in a KC, in which a 1 s odor

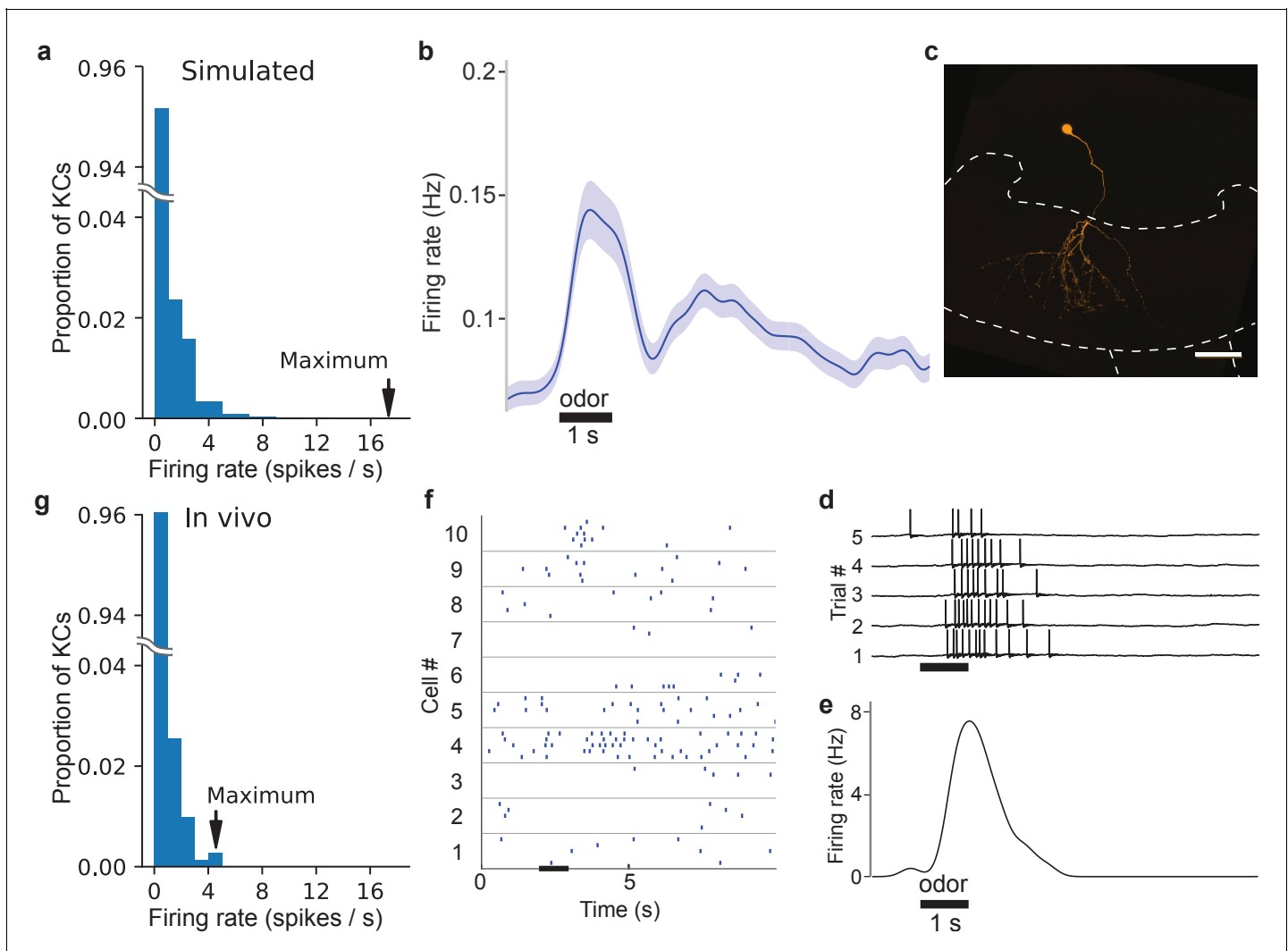

**Figure 6.** In vivo, some KC responses are hyperactive, as predicted by simulations. (**a**) Distribution of KC firing rates in simulation producing realistic GGN depolarization. (**b**) KC firing rate averaged over 707 KC-odor pairs. Shaded region indicates standard error of the mean. Black horizontal bar: 1 s odor stimulation. (**c**) A hyperactive response KC filled with dye (mushroom body calyx and pedunculus outlined with dashed lines (scale bar: 50 μm) and (**d**) its recorded membrane potential in response to odor stimulus, and (**e**) its average firing rate elicited by this stimulus computed in a 100 ms window. (**f**) Sample KC spike trains. (**g**) Histogram of average firing rate of KCs across trials upon odor presentation measured in a 1.4 s window. The last bin in the rightward tail includes two different KCs.

The online version of this article includes the following figure supplement(s) for figure 6:

**Figure supplement 1.** Example distributions of KC spike counts in four instantiations of the network architecture in *Figure 6*.

pulse elicited an average of 7 spikes (*Figure 6d and e*). *Figure 6f* shows spike times obtained from 10 other KC-odor pairs. Overall, the distribution of spike counts in KCs we tested in vivo was clustered close to 0 but included a long rightward tail (*Figure 6g*), in agreement with our model (*Figure 6a*). KCs generating especially strong responses were not located in any particular part of the calyx, nor did they feature distinguishing morphologies (data not shown).

Although relatively rare KC odor responses with many spikes have been observed before (*Perez-Orive et al., 2002*; *Gupta and Stopfer, 2014*), the origin and significance of these responses were unclear. We tested this with a series of simulations in which we systematically reduced the number of high spike-rate KCs by disconnecting them from the network. We found that when KCs producing more than five odor-elicited spikes were removed (*Figure 7—figure supplement 1*), a different set of KCs emerged to replace them by becoming more active. However, the replacement high spike-rate KCs were fewer in number than the original high-firing set (*Figure 7a*). As a result, the population spike time histogram and corresponding GGN voltage deflections became smaller (*Figure 7b and c*). As we reiterated this process, deleting newly-emerged high spike-rate KCs from the network with each round, the rightward tail of the KC population's spike count distribution shortened, the number of low-spike KCs increased (*Figure 7a*), and, overall, the total number of spikes produced by the entire KC population decreased monotonically. Eventually, when no KCs in the reiterative simulation generated six or more odor-elicited spikes, we decreased the spike limit, first to three or more spikes, and then, to any spikes (*Figure 7—figure supplement 1*). Throughout this process, GGN's odor-elicited response gradually decreased, shifting from extended depolarization to unrealistic, isolated peaks. Ultimately, no spiking KCs, or responses in GGN, remained (*Figure 7c*). Our analysis showed two things: first, the high spike-rate KCs in our model are not a static population, but rather change in identity, reflecting emergent features of the network's distribution of synaptic conductances, PN input, and GGN feedback; and second, high-spike rate KCs augment the depolarization of GGN to realistic levels.

## Odor-evoked hyperpolarization in GGN originates in IG spiking

Our intracellular recordings from GGN revealed extended periods of odor-elicited hyperpolarization (*Figure 4*), something not previously reported in recordings made in vivo nor explained by existing models of GGN within its olfactory network. We hypothesized that these periods of hyperpolarization might originate in the activity of IG, a neuron known to share reciprocal inhibition with GGN (*Figure 1a*, right) (*Papadopoulou et al., 2011*). Specifically, we hypothesized that an increase in IG activity might underlie the periods of hyperpolarization in GGN, with IG's activity increase following release from inhibition from GGN. To test this, we first tried adding IG to our model according to the simple reciprocal connectivity plan shown in *Figure 1a*. We tested a range of synaptic strengths and time constants for GGN→IG, and IG→GGN synapses, and manipulated IG's baseline firing rates, but none of these model variations could generate odor-elicited hyperpolarization in GGN (data not shown). Thus, our model suggested simple reciprocal interactions between these neurons cannot generate this response.

We sought to test this in vivo. We reasoned that if spiking in IG increased solely because inhibition from GGN decreased, then IG spiking should only increase after odor stimuli ended and activity in GGN returned to baseline. The location and most properties of IG are unknown, and we were not able to identify it in our recordings, making it difficult to directly test our idea. However, a previous report showed spikes in IG correlate one-to-one with IPSPs in GGN (*Papadopoulou et al., 2011*), suggesting we could infer IG's activity and properties by examining GGN's membrane potential. Our recordings made in vivo from GGN revealed IPSPs in the absence of odors, and that odor presentations caused the frequency of IPSPs to increase (Wilcoxon signed-rank test, N = 198 pairs, W = 2328.5, p<<0.001; responses from two animals are shown in *Figure 8a*, and responses from 1257 trials with several odors from 47 GGNs are shown as a peri-stimulus time histogram in *Figure 8b and c*.) Assuming these IPSPs originate as spontaneous and odor-elicited spikes in IG (*Papadopoulou et al., 2011*), these results show that IG's responses to an odor pulse are delayed relative to the odor onset, and lengthy. Notably, as evident in *Figure 8b*, IG's firing rate begins to increase before GGN's membrane potential returns to baseline. This result rules out the possibility that IG's odor response is driven solely by a reduction in inhibition from GGN, and suggests, rather, that IG receives odor-driven excitation from another pathway. It has been hypothesized that IG receives input from KCs (*Papadopoulou et al., 2011*). Using our model, we could indeed reproduce

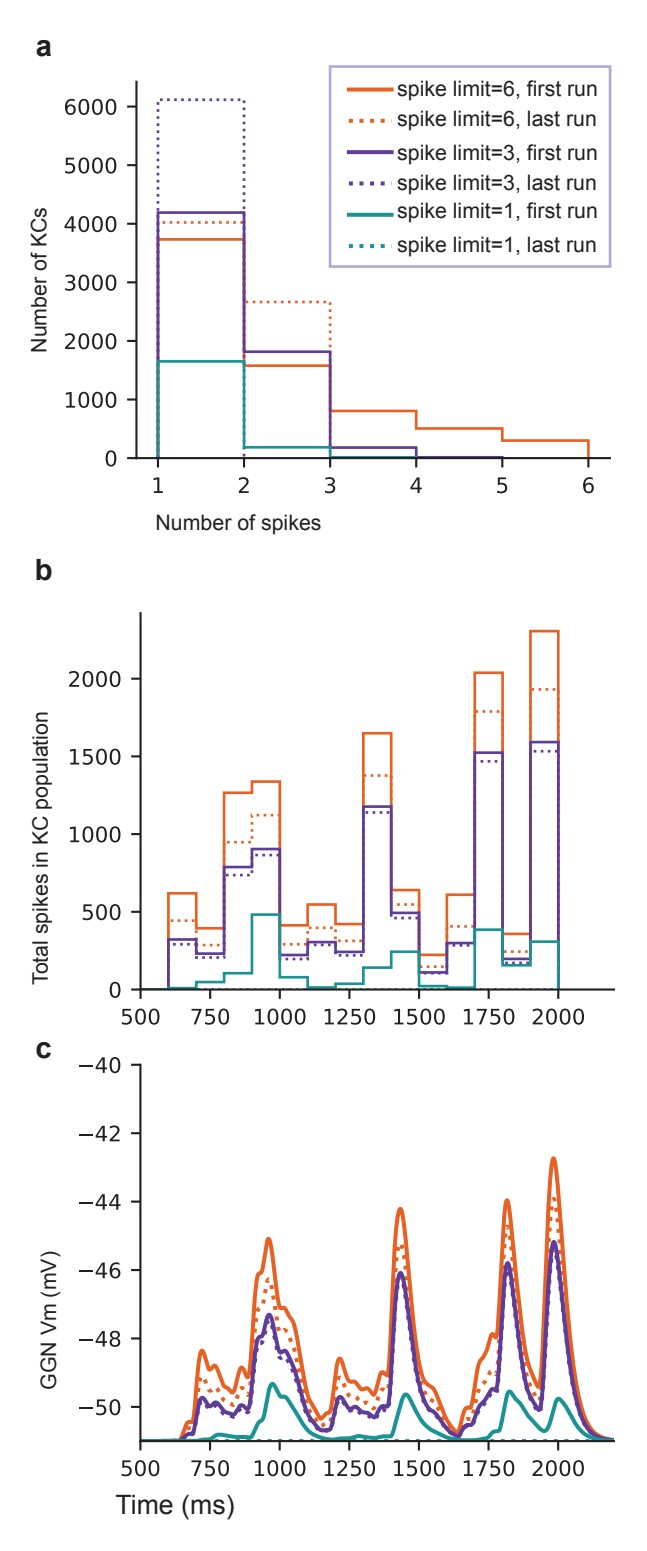

**Figure 7.** High spike-rate KCs contribute to sustaining GGN's odor-elicited depolarization. (**a**) Histogram of spike counts in KCs for one template network at the first and the last run of each spike count limit. (**b**) Population spike time histogram of KCs (100 ms bins) for the same simulations. (**c**) GGN membrane potential. See text for the procedure. Legend is same for **a–c**.

The online version of this article includes the following figure supplement(s) for figure 7:

**Figure supplement 1.** Flow chart of the reiterative simulation of the same model while removing high-firing KCs.

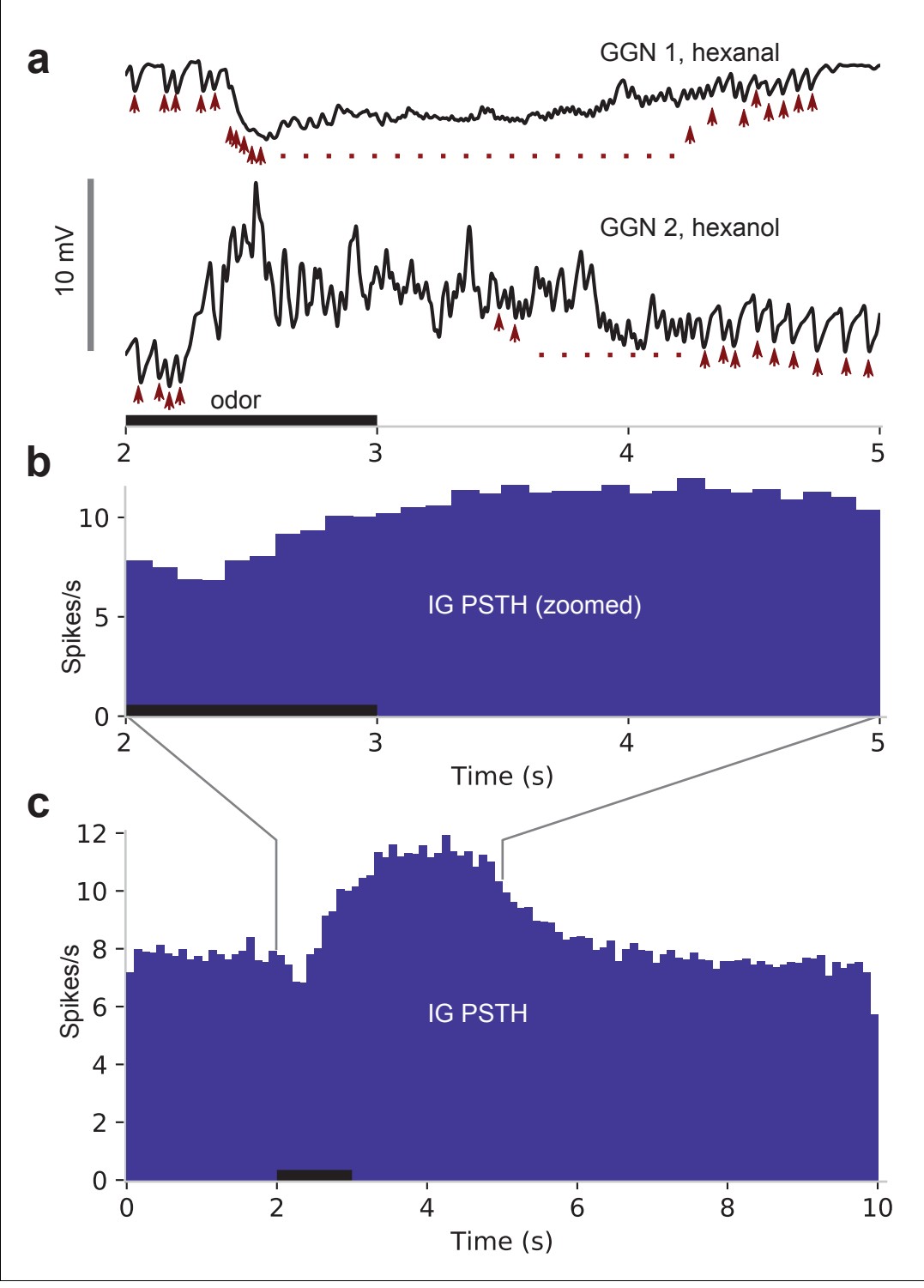

**Figure 8.** IPSPs in GGN suggest IG spikes upon odor presentation and is responsible for hyperpolarizations in GGN. (a) In vivo recordings of GGN's membrane potential from two animals showing IPSPs (arrowheads) believed to originate as spikes in IG. Vertical scale bar: 10 mV. (b) Peristimulus time histogram (PSTH) shows IPSP peak times from 1257 odor-trials across 47 GGNs, presumably reflecting spikes in IG. Because IG begins to spike before GGN's membrane potential returns to baseline, IG's odor response cannot be driven by disinhibition from GGN. Black horizontal bar: odor presentation in panels (a) and (b), which share the time axis. (c) Same as panel (b) but showing full responses.

realistic hyperpolarization in GGN's membrane potential by adding excitatory synapses to IG from either the KCs or the PNs (*Figure 9a–e*). Our simulations produced a variety of membrane potential profiles in GGN including hyperpolarizations; the profiles varied with the odor-driven activity patterns in PNs (*Figure 9f–i*).

Our analysis of IG's activity suggested that it spikes in the absence of odor stimulation (*Figure 9j*, top black traces). Thus, a remaining question concerned the source of excitation driving this spontaneous activity. KCs are nearly silent when odors are not present (*Perez-Orive et al., 2002*; *Gupta and Stopfer, 2014*), making it unlikely that they provide the sole source of excitation to IG. PNs, though, spike spontaneously because they receive direct, powerful input from spontaneously active ORNs (*Joseph et al., 2012*), suggesting a PN-driven pathway might be responsible for spontaneous activity in IG. To test this in vivo, we completely silenced PNs and KCs by bilaterally cutting the animal's antennal and labral nerves (*Joseph et al., 2012*) and then recording intracellularly from GGN. We found that spontaneous IPSPs in GGN persisted in preparations with silenced PNs and KCs (*Figure 9j*, bottom red traces), demonstrating that IG's spontaneous spiking either arises intrinsically or is driven by a source other than PNs or KCs.

## Discussion

Inhibitory neurons play critical roles in regulating and shaping olfactory responses in vertebrates and invertebrates (*Kay and Stopfer, 2006*). In insects, these roles are performed by relatively few neurons that can be interrogated efficiently, revealing fundamental principles of olfactory coding. The unique giant GABAergic neuron GGN plays a central role in structuring olfactory codes in the locust mushroom body by regulating the excitability of KCs and parsing their responses into rhythmic bursts. We combined intracellular recordings from GGN and KCs, and developed a new morphologically detailed model of GGN as a focus of analysis to investigate GGN's properties, inputs, and outputs. Further, we used a broader model of the olfactory system built around GGN to explore several basic properties of the olfactory network. Our computational model, based on our electrophysiological recordings, successfully reproduced the sparse activity of KCs and the membrane dynamics of GGN in the locust brain while providing new hypotheses about how the mushroom body circuit processes odor information.

It is clear that GGN inhibits KCs, but the pathway carrying this inhibitory signal through GGN is uncertain. It has been proposed that signals generated within GGN's α lobe branch propagate to its distant calyceal branch, where they provide global inhibition to all KCs (*Papadopoulou et al., 2011*). But, the enormous size, extensive branching, and passive conduction of GGN brought us to hypothesize that signals arising in GGN's α lobe branch would attenuate to such an extent as they travel to the calyx that they would be unable to effectively inhibit KCs. If this hypothesis were correct, GGN's inhibition of KCs would presumably occur by a different mechanism, such as through local inhibition entirely within the mushroom body calyx. Non-spiking interneurons in insects are often large with complex splays of neurites in separate brain areas, suggesting their far-flung branches may be functionally isolated, serving separate local computations (*Burrows, 1981*). A variety of complex, local processing has been proposed to occur within the locust mushroom body calyx (*Leitch and Laurent, 1996*). Our simulation of GGN's morphological and electrical properties shows that realistic, odor-elicited levels of depolarizations of ~10 mV in GGN's α lobe branch do indeed attenuate greatly with distance to amplitudes as low as 5 mV in parts of the calyx. However, earlier studies of non-spiking neurons in invertebrates showed depolarizations of this amplitude could evoke enough neurotransmitter release to affect responses of postsynaptic neurons. For example, (*Burrows and Siegler, 1978*) showed that depolarizations of only about 2 mV in a non-spiking interneuron in the metathoracic ganglion of the locust can change the firing rate of its postsynaptic motor neuron. Similarly, (*Manor et al., 1997*) showed in a graded synapse in the lobster stomatogastric ganglion that voltage steps from $-50$ mV to $-45$ mV can reliably evoke postsynaptic effects. Thus, we conclude that GGN has the biophysical capacity to convey signals from KCs at GGN's α lobe branches to the calyx, providing effective global inhibition to all KCs.

GGN's arborizations in the calyx extend to different lengths, suggesting signals arising in the α lobe could attenuate more in some of its calyceal branches than in others. Our simulations indeed showed the amount of depolarization reaching GGN's distant branches varied with their locations, but only by a few millivolts (*Figure 2b*), within the range of variation caused by other factors, such as

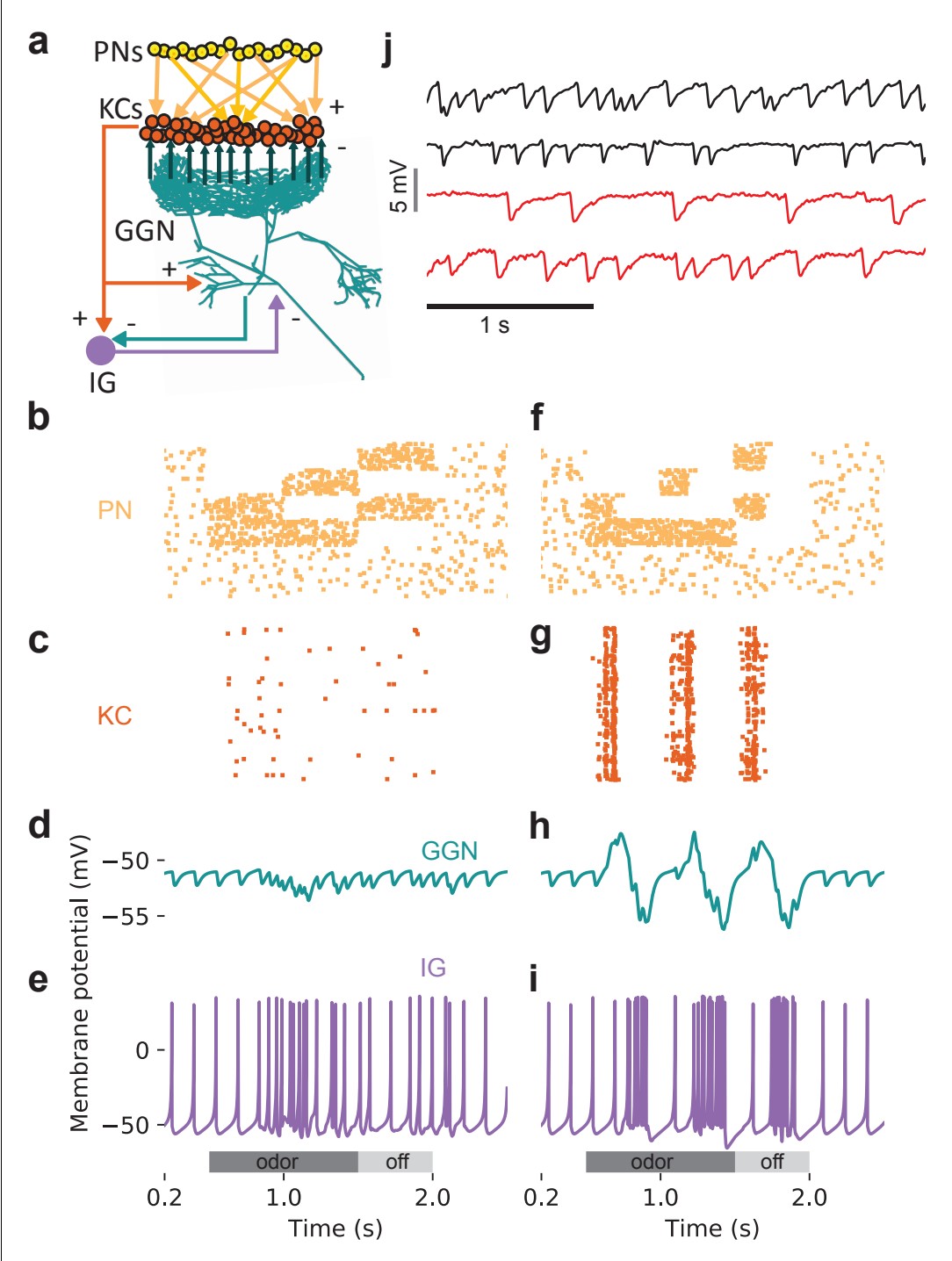

**Figure 9.** An additional, unidentified olfactory pathway to IG is needed to explain odor-elicited hyperpolarization in GGN. (**a**) Schematic of model with IG receiving direct excitation from KCs and reciprocal inhibition from GGN. (**b**) Spike raster of PN activity producing GGN hyperpolarization. (**c**) Spike raster of KC population activity in the same simulation. (**d**) Simulated GGN membrane potential including odor-elicited hyperpolarization mimics responses observed in vivo (e.g. *Figure 4*, Animal 3, hexanal). (**e**) Corresponding simulated IG membrane potential. This simulation included a 200 ms synaptic delay from KCs to IG. (**f–i**) Varying the temporal pattern of PN population activity can produce different response pattern in the same network as e. (**f**) Raster plot of PN activity with a different temporal pattern from that in e. (**g**) Spike raster showing concomitant KC activity. (**h**) Simulated GGN membrane potential including odor evoked de- and hyperpolarization (similar to *Figure 4*, Animal one hexanal). (**i**) Corresponding simulated IG membrane potential. (**j**) Spontaneous activity in IG does not originate in the antennal lobe. Top two black traces show spontaneous IPSPs in GGN's

*Figure 9 continued on next page*

*Figure 9 continued*

membrane voltage from two animals with intact olfactory systems. The bottom two red traces are from the left and the right GGN in another animal in which the antennal lobes had been silenced by cutting both antennal nerves. Vertical scale bar 5 mV, horizontal scale bar 1 s.

synaptic strength. Consistent with this, imaging experiments in APL, the *Drosophila* analog of GGN (*Lin et al., 2014*) revealed different levels of Ca$^{2+}$ activity in different spatial regions of its arbor when the animal was stimulated with low concentrations of odors (*Inada et al., 2017*). Inada et al. also found that some KCs are more effective than others at activating APL, and that APL, in turn, provides diverse levels of inhibition to different subsets of KCs in the same brain. By optogenetically activating a small subset of PNs, Inada et al. showed that APL could provide local inhibition to a subset of KCs, likely through dendrodendritic connections between KCs and APL within the calyx. Indeed, reconstructions of APL based on serial section electron microscope images show that it receives synaptic connections from KCs throughout its arbor, including within the calyx (*Eichler et al., 2017*; *Zheng et al., 2018*).

Could local and global inhibition of KCs operate in parallel? In vivo, strong odor stimuli appear to drive both GGN and APL to provide global inhibition to all KCs (*Papadopoulou et al., 2011*; *Lin et al., 2014*), and our model shows (*Figure 2*) that depolarizations of GGN evoked by strong input can spread robustly throughout the giant neuron's arbor. We speculate that in the locust, weaker stimuli might evoke weaker depolarization in GGN's α lobe branch, leading to attenuated signals in GGN's distal branches too weak to inhibit KCs. If, GGN, like APL, forms reciprocal synapses with KCs within the calyx, it might be able to inhibit KCs there locally even in the absence of a global inhibitory signal. In this scenario, a KC driven to spike by a weak odor stimulus would effectively inhibit only its close neighbors via local interactions with GGN. Assuming randomly-connected input from PNs to KCs (*Jortner et al., 2007*; *Caron et al., 2013*; *Eichler et al., 2017*), we can speculate about possible impact of strong or weak odor stimuli on the circuit. Strong odor stimuli might drive this circuitry into a global, population-wide, winner-take-all scenario in which only KCs receiving the strongest inputs manage to spike. Weak stimuli, on the other hand, might recruit only local inhibition, allowing a distribution of winners to emerge locally, resulting in a center-surround-type of contrast enhancement.

Inhibition from GGN is known to sparsen the firing of KCs and to impose rhythmic time windows on their responses (*Papadopoulou et al., 2011*; *Gupta and Stopfer, 2012*). Notably, our model also predicts that feedback inhibition from GGN can expand the dynamic range of inputs able to activate KCs (*Figure 3*). This prediction is consistent with observations made by imaging calcium in *Drosophila* where blocking inhibitory inputs to KCs narrowed the dynamic range of their odor responses (*Inada et al., 2017*). Why might expanding the dynamic range of KCs benefit olfactory coding? In the brain, a wide range of synaptic strengths exists even within a given type of neuron, and, further, synaptic strength can change over time and with experience (reviewed in *Barbour et al., 2007*). As in other systems, it is possible that homeostatic mechanisms tune conductances in KCs to match their inputs, enabling them to respond appropriately to a broad and changing range of inputs (reviewed in *Marder and Goaillard, 2006*). Our results suggest an additional mechanism: feedback inhibition from GGN that helps KCs function robustly by expanding their sensitivities to a wider range of inputs, enabling them to generate consistent responses.

Although KCs generally respond very sparsely to any given odor, with very few cells firing just one or a few spikes (*Laurent and Naraghi, 1994b*; *Perez-Orive et al., 2004*; *Mazor and Laurent, 2005*), examples of KCs that fire substantially more spikes, spontaneously and in response to odors, have been reported (*Perez-Orive et al., 2002*; *Gupta and Stopfer, 2014*). Notably, our simulations that gave rise to realistic, sustained responses in GGN always included some high-spike rate KCs. When we removed these high-firing rate KCs from the network, new high-firing rate KCs arose to replace them (*Figure 7*). This analysis showed that high-firing rate responses in KCs are an emergent property of the olfactory network, determined by the balance of input KCs receive: distributed patterns of excitation from PNs; and rhythmic inhibitory feedback from GGN. Deleting all high spike-rate KCs from a network had a modest effect on GGN, slightly reducing the amplitude of its sustained, odor-elicited depolarization. Deleting KCs generating 3–6 spikes per stimulus reduced GGN's depolarization to unrealistic levels, leaving only widely-separated peaks of activity (*Figure 7*).

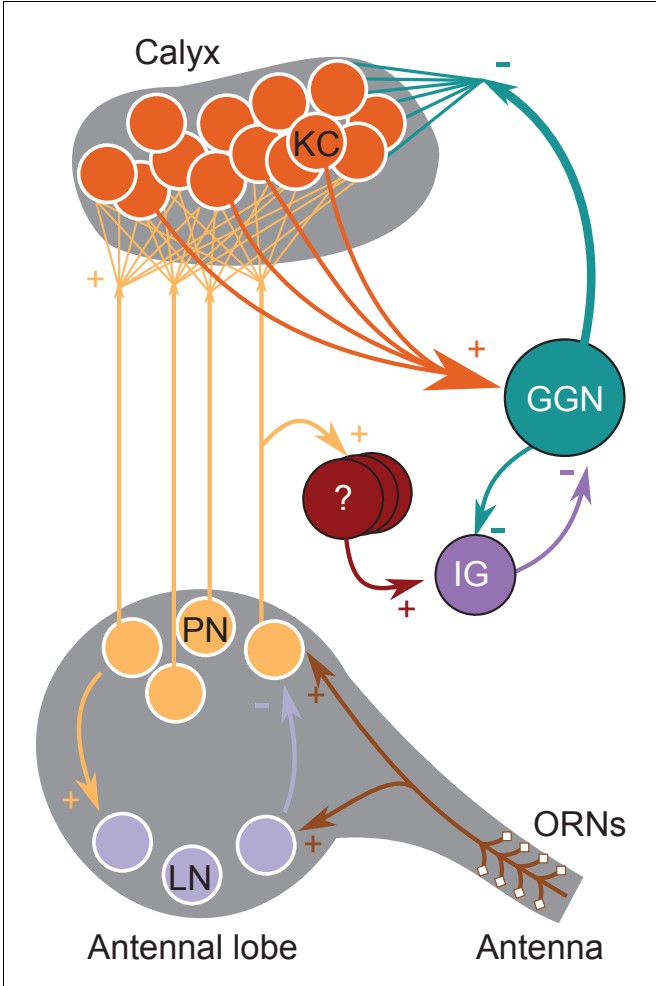

**Figure 10.** Updated olfactory connectivity model where odor responses in the KC population are gated by feedback inhibition from GGN and inhibition of GGN via odor evoked spiking in IG, which itself receives input from an unknown olfactory pathway.

Our intracellular recordings from GGN revealed more elaborate membrane potential dynamics than previously reported, including reliable and prolonged stimulus-dependent periods of hyperpolarization, that varied with odor and animal. In our model, simple reciprocal inhibition of GGN by the inhibitory neuron IG could not reproduce these features. But we found introducing odor-elicited excitatory input to the inhibitory neuron IG could give rise to hyperpolarizations of GGN's membrane potential (*Figure 10*). The direct source of odor-elicited excitatory drive to IG is unknown, but could, in principle, be traced to KCs. Indeed, a version of our model in which all KCs synapse upon IG reliably reproduced fairly realistic odor-elicited hyperpolarizations in GGN. This result leads us to hypothesize that KCs, which require strong and coincident input from PNs to spike (*Perez-Orive et al., 2002*; *Jortner et al., 2007*) may serve as thresholding elements in an odor pathway linking the antennal lobe to IG, transferring odor-evoked activity, but not lower levels of spontaneous activity, from PNs. The identity of this proposed odor pathway is not known.

Why are hyperpolarizations of GGN elicited by only some odors in some animals, as we saw in vivo (*Figure 4*)? Our simulations suggest that odor-specific patterns of spiking across the PN population (*Stopfer et al., 2003*) may underlie this variability (*Figure 9b–i*). Another possibility, not tested here, is that different sets of KCs could drive GGN and IG differentially such that odor-driven KCs strongly activate IG, but only weakly activate GGN, causing GGN to hyperpolarize. Different classes of KCs have been reported to differentially activate APL in *Drosophila* (*Inada et al., 2017*), but this possibility remains unexplored in locust.

Our results indirectly suggest IG is active in the absence of odor stimuli (*Figures 8* and *9*). IG may generate this spontaneous activity on its own or may inherit it from other olfactory or non-olfactory neurons. KCs are nearly silent in the absence of odor stimuli, making them unlikely direct sources of this activity, but it is possible that even extremely sparse spontaneous activity distributed over the population of 50,000 KCs could suffice to drive spontaneous spiking in IG. This possibility would be difficult to test in vivo.

Several caveats apply to predictions of our model. Although we explored a wide range of values for GGN's membrane and cytoplasmic conductances to test our model for robustness (*Figure 2g*), we cannot exclude the possibility that ion channels are distributed unevenly across GGN, potentially allowing parts of the neuron to behave in ways we could not observe in our recordings. We simplified our models of KCs as single compartments, and in some cases collapsed multiple synapses into a single equivalent synapse; however, these simplifications should not affect our predictions in a qualitative sense. Finally, our model excludes plasticity mechanisms, so our predictions apply best to odors that are novel or delivered at long intervals.

In sum, we used biophysically detailed simulations in combination with electrophysiology performed in vivo to explore the olfactory circuit of the locust mushroom body. Our intracellular and patch recordings revealed new features of GGN, a neuron that plays a central role in shaping olfactory responses, and of the KC population. These results extend our understanding of the olfactory system, highlighting ways different components interact, and provide new constraints and predictions for additional research.

# Materials and methods

**Key resources table**

| Reagent type (species) or resource | Designation | Source or reference | Identifiers | Additional information |
|---|---|---|---|---|
| Software, algorithm | NEURON | https://neuron.yale.edu/neuron/ | RRID:SCR_005393 | Versions 7.4 and 7.5 with Python 2.7 |
| Software, algorithm | NeuroLucida 360 | https://www.mbfbioscience.com/neurolucida360 | RRID:SCR_016788 | |
| Chemical compound, drug | Neurobiotin | Vector Laboratories | SP-1120 | 1–5% in 2–3M potassium acetate |
| Chemical compound, drug | Avidin-Alexa complex | Thermo Fisher Scientific | S11226 | 0.01 mg/ml in 1X PBS |

## Dissection and electrophysiology

Newly eclosed adult locusts of both sexes picked randomly from our crowded colony were immobilized and the brain was exposed, desheathed and superfused with locust saline as described before (*Brown et al., 2005*). Our dissection included cutting the labral nerves and removing the suboesophageal ganglion (SOG), precluding stimulation via olfactory receptors in the palps. We usually obtained recordings from one GGN per animal and observed no systematic variation in results over the course of about a year. GGN is not visible from the brain's surface. To record from GGN, a sharp glass micropipette filled with 2 or 3M potassium acetate with 5% neurobiotin was inserted into the peduncle region of the mushroom body; when impaled, GGN could be identified by its characteristic pattern of IPSPs in the voltage record (*Figure 9j*). At the end of the recording session, neurobiotin was injected into the cell iontophoretically using 0.2 nA current pulses at 3 Hz for 10 to 20 min, allowing the cell's identity to be confirmed by subsequent imaging.

To test whether IPSPs in GGN originate from spontaneous activity in PNs, we first silenced the PNs by cutting both antennal nerves at the base (*Joseph et al., 2012*) and then searched for GGN, which could be identified by its ongoing unique pattern of IPSPs and confirmed by its morphology, revealed by subsequent filling with neurobiotin and imaging.

For patch clamp recordings from KCs, the initial dissection was performed as described above. Patch pipettes were pulled to between 7 and 12 MΩ, filled with locust internal solution (*Laurent et al., 1993*) as well as a neural tracer for subsequent histology (either 12 mM neurobiotin for later conjugation with an Avidin-Alexa complex, or 20 μM Alexa Fluor tracer with absorption wavelengths of 488, 568 or 633). Patch recordings were made in current clamp mode,

and data were only analyzed if the observed membrane potential was within the previously reported range for KCs (−55 to −65 mV) and if either LFP or membrane potential oscillations were observed in response to odor stimulation. Firing rates were obtained by smoothing the PSTH with a 100 ms SD Gaussian window.

## Stimulus delivery

For GGN recordings, odor pulses were delivered to the ipsilateral antenna as described in *Gupta and Stopfer (2012)*. The odorants included 1-hexanol at the dilution of 1% v/v, 1-hexanol, hexanal, methyl benzoate, benzaldehyde, and cyclohexanone mixed at the dilution of 10% v/v in mineral oil, and 100% mineral oil.

For KC recordings, the following odors were delivered similarly: 1-hexanol, hexanal, cyclohexanol, 1-octananol, citral, geraniol, ethyl butyrate, 1-butanol, benzaldehyde, eugenol, 3-methyl-2-butenol, methyl jasmonate, decanal, methyl salicylate, linalool, limonene, pentyl acetate, (all 10% v/v in mineral oil, except methyl salicylate and limonene, at 40% each) and 100% mineral oil.

## Histology and immunostaining

Brains were dissected from the head capsule and fixed in 4% paraformaldehyde overnight, then conjugated with Avidin-Alexa 568 or Avidin-Alexa 488. Some brains (*Figure 1a*) were first immunostained with mouse nc82 primary antibody (DSHB Cat# nc82, RRID:AB_2314866, deposited to the DSHB by Buchner, E.) and Alexa 568 conjugated anti-mouse IgG secondary antibody (*Shimizu and Stopfer, 2017*).

## Imaging and neuronal tracing

Brains were dehydrated in an ethanol series and cleared with methyl salicylate or CUBIC (*Susaki et al., 2014*), mounted in methyl salicylate or mineral oil respectively, and imaged with a Zeiss LSM 710 confocal microscope. Two GGNs were traced in detail from 3D image stacks using NeuroLucida 360 software (MBF Bioscience, Williston, Vermont). The traces were converted to SWC format for further processing and cleanup using NLMorphologyConverter (www.neuronland.org). The two traces were very similar, and models based on both gave similar results. Here, we show results obtained from the GGN imaged at higher resolution.

## Statistics

For each of 198 GGN-odor pairs, average IPSP rates were calculated over five trials in windows beginning 2 s before, and 2 s after, odor presentations. Wilcoxon-signed rank tests from the scipy package were used to compute the statistic and the two-sided *p*-value.

## Computational model

GGN morphologies in SWC format were imported into NEURON and converted into passive models in NEURON's hoc format. The resting membrane potential was set to −51 mV, as observed in vivo. The single compartmental KC model reported by *Wüstenberg et al. (2004)* was translated manually into a NEURON model. This Hodgkin-Huxley type model included fast and slow Na+ channels, and delayed rectifier, transient A type, and slow transient outward K+ channels. The passive reversal potential of the KCs was set to −70 mV. Custom Python scripts were written to set up network models and simulation experiments using NEURON's Python interface. The simulations were run on NIH's Biowulf supercomputer cluster (http://hpc.nih.gov) and simulation results were saved in HDF5 based NSDF format (*Ray et al., 2016*) and later analyzed with custom Python scripts.

## PN activity model with a fixed responsive population

Modeled PN spike train rates were based on firing statistics reported in Figure 2c–e in *Mazor and Laurent (2005)* from which we infer 77% of PNs are spontaneously active. Odor presentations were set to evoke spiking in ~20% of PNs, each spiking at an average rate of 20 Hz (*Mazor and Laurent, 2005*; *Jortner et al., 2007*). A 20 Hz sinusoid with amplitude 0.4 times the average spiking rate was further superimposed on odor-elicited spiking to model oscillatory activity generated in the antennal lobe. Based on our own observations in vivo we set 10% of spontaneously active PNs to respond

with inhibition to odor presentations. We used a non-homogeneous Poisson generator to create the spike trains based on these rates.

## PN activity model with a shifting responsive population

To test the significance of varying temporal structure in PN firing patterns, we assumed ~30% of the PNs were unresponsive to a given odor, and divided the other 70% into four equally sized groups with the following odor-elicited sequences of excitation (E) and inhibition (I): EEI, EIE, IEI and IIE, with the last epoch occurring upon stimulus offset (*Laurent and Davidowitz, 1994a*). Within each group, excitation epochs featured shifting sets of active PNs, with new PNs recruited during each LFP cycle (*Mazor and Laurent, 2005*). Each group started its excitatory epoch with activation of 70% of its members and 10% were recruited in each of the next three LFP cycles. In this scheme at most ~30% of PNs were active at any given time during odor presentations.

## Connectivity from GGN to KCs

Each KC received one graded inhibitory synaptic input from a randomly assigned point on GGN's calyceal branches. The strength of each synapse was adjusted to keep the KC's membrane potential close to −60 mV when bombarded by spontaneous activity from PNs, as observed in vivo (*Joseph et al., 2012*). The graded synapse was modeled as a NEURON mechanism based on published descriptions (*Manor et al., 1997*; *Papadopoulou et al., 2011*). In some simulations the individual synaptic conductances were selected from a lognormal distribution with the mean adjusted to produce realistic KC activity, that is to evoke spiking in about 10% of the KCs with most of them responding with fewer than five spikes to odor stimuli. The standard deviation of the lognormal distribution had the same value as the mean, based roughly on *Song et al. (2005)*. In network simulations we considered KC activity to be realistic if ~10% or fewer KCs spiked in response to odor stimulus. Also, we considered KC activity to be unrealistically low if it failed to depolarize GGN.

## Connectivity from PNs to KCs

Half of the PN population was randomly and independently selected as presynaptic partners for each KC. If two subsets of size $m$ and $n$ are randomly and independently selected from a set of size $q$, the expected size of their intersection is $s = m * n/q$. Thus, with 800 PNs and each KC receiving input from 400 PNs, the expected number of PNs shared by any two KCs is 400 * 400/800 = 200, that is they share about 50% of their presynaptic PNs, as shown in vivo (*Jortner et al., 2007*). We carried out a parameter search to obtain a preliminary estimate of a PN→KC synaptic conductance sufficient to evoke a few spikes in a single KC model when driven with the expected combined PN firing rate during odor stimuli. Later, in the full network model, we tested values around this conductance along with different GGN→KC conductances. For heterogeneous conductance models, the conductances of individual synapses were selected from a lognormal distribution with the mean adjusted to produce realistic KC activity, as described above.

To test whether diverse synaptic delays could alleviate synchronized spiking in KC population, we created a set of networks where PN→KC synapses were assigned a range of delays from a normal distribution with mean 7.5 ms and standard deviation 3 ms. We chose these values with the animal's physiology in mind, by assuming a conservative speed of 0.2 m/s = 200 μm/ms for signal propagation through dendrites (*Larkum et al., 1996*). The length of PN axon is about 1500 μm, so the total time delay from PN to KCs should be under 1500/200 = 7.5 ms, consistent with the delay of 6 ms observed in vivo by *Jortner et al. (2007)*. The maximum diameter of a large KC dendritic arbor is about 600 μm in the calyx and thus, the longest delay between two KCs receiving spikes from the same PN in their soma should be about 3 ms.

## IG model and connectivity

To simulate IG, we used a single compartmental Izhikevich-type model of a regular spiking (RS) pyramidal cell from the Model DB repository (https://github.com/ModelDBRepository/39948) modified to include graded synaptic input from GGN. To model IG's spontaneous firing, sufficient current was injected into it to generate ~7 spikes/s. A single inhibitory synapse with −80 mV reversal potential connected IG to one of GGN's basal dendrite segments. The strength and time constant for this synapse were adjusted to produce IPSP amplitudes matching those observed in vivo. The same GGN

segment was connected to IG via a graded synapse. In some simulations either the output of all PNs or the output of all KCs were connected to IG via excitatory synapses. The synaptic weights from KCs to IG were selected from a lognormal distribution.

### Data analysis

Most analyses and 3D visualizations were carried out with custom Python scripts using published modules including numpy, scipy, networkx, matplotlib, h5py, pandas and scikitslearn.

Analyses of patch clamp recordings from KCs were carried out with custom MATLAB scripts.

## Acknowledgements

We thank Tianming Li and Ian McBain for help with tracing GGN morphology from confocal image stacks, Nitin Gupta (IIT Kanpur, India) for providing a confocal image stack of GGN for a pilot model and helpful advice for making recordings from GGN. We thank the members of the Stopfer lab for feedback and suggestions and Diantao Sun and Kui Sun for excellent animal care. We also thank Vincent Schram and Lynn Holzclaw of the NICHD Microscopy and Imaging Core for help with confocal imaging, George Dold and Bruce Pritchard of NIMH Section on Instrumentation for help with electrophysiology data acquisition setup, and Theodor Usdin, NIMH/NIH, for suggestions on tissue clearing and for providing initial reagents for the CUBIC protocol. We thank the Developmental Studies Hybridoma Bank for the nc82 antibody. This work utilized the computational resources of the NIH HPC Biowulf cluster (http://hpc.nih.gov). This work was funded by an intramural grant from NIH-NICHD to MS.

## Additional information

### Funding

| Funder | Grant reference number | Author |
|---|---|---|
| Eunice Kennedy Shriver National Institute of Child Health and Human Development | Intramural grant | Subhasis Ray<br>Zane N Aldworth<br>Mark A Stopfer |

The funders had no role in study design, data collection and interpretation, or the decision to submit the work for publication.

### Author contributions

Subhasis Ray, Conceptualization, Data curation, Software, Formal analysis, Investigation, Visualization, Methodology; Zane N Aldworth, Data curation, Formal analysis, Investigation, Visualization, Methodology; Mark A Stopfer, Conceptualization, Resources, Supervision, Funding acquisition

### Author ORCIDs

Subhasis Ray (iD) https://orcid.org/0000-0003-2566-7146
Zane N Aldworth (iD) https://orcid.org/0000-0002-0647-8465
Mark A Stopfer (iD) https://orcid.org/0000-0001-9200-1884

### Decision letter and Author response

Decision letter https://doi.org/10.7554/eLife.53281.sa1
Author response https://doi.org/10.7554/eLife.53281.sa2

## Additional files

### Supplementary files
• Transparent reporting form

## Data availability

Morphology traces have been deposited to neuromorpho.org (http://neuromorpho.org/dableFiles/stopfer/Supplementary/GGN_20170309_sc.zip). Identical morphology data along with computational models and simulation scripts are available on GitHub (https://github.com/subhacom/mbnet; copy archived at https://github.com/elifesciences-publications/mbnet). Model source code curated by Model DB is available under the accession number 262670.

The following dataset was generated:

| Author(s) | Year | Dataset title | Dataset URL | Database and Identifier |
|---|---|---|---|---|
| Ray S, Aldworth Z, Stopfer M | 2020 | Locust olfactory network with GGN and full KC population in the mushroom body | http://modeldb.yale.edu/262670 | ModelDB, 262670 |

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
