## [Decision Letter]

**Acceptance summary:**

This study uses a combination of electrophysiological recordings and modeling to examine how a wide-field inhibitory neuron contributes to the dynamics of odor responses. The work argues that heterogeneous excitation from a direct olfactory pathway as well as odor-driven inhibition both sculpt the response properties of the wide-field inhibitory neuron and contribute to odor coding. The combination of modeling and recordings provides new insights and testable hypotheses about how inhibition may expand the dynamic range of odor detection.

**Decision letter after peer review:**

[Editors’ note: the authors submitted for reconsideration following the decision after peer review. What follows is the decision letter after the first round of review.]

Thank you for submitting your work entitled "Feedback inhibition and its control in an insect olfactory circuit" for consideration by *eLife*. Your article has been reviewed by three peer reviewers, and the evaluation has been overseen by Kristin Scott as Reviewing Editor and a Senior Editor. The reviewers have opted to remain anonymous.

Our decision has been reached after consultation between the reviewers. Based on these discussions and the individual reviews below, we regret to inform you that your work will not be considered further for publication in *eLife* at this time. The reviewers were positive about the significance of the study but had several concerns, detailed in individual reviews below. While many of the concerns may be addressed with changes to the text, one experiment regarding testing the model in Figure 3 seems likely to require additional time to complete beyond a two month time window allowed for revision. However, the reviewers do feel the work could be made acceptable after the reviewers' concerns are addressed. Should you choose to submit a new version of this work, we will endeavor to have it reviewed by the same BRE member and reviewers.

Reviewer #1:

Overall I like this combination of modelling and experiments – it covers an important topic and reaches interesting conclusions. I have some concerns that the authors should address before publication.

Ray et al. construct a morphological model of the locust's giant GABAergic neuron (GGN) based on its reconstructed anatomy, and use it to model the spread of depolarization within GGN. The model predicts that although GGN is very large and doesn't spike, signals should still spread readily from one part to the other, though they get attenuated. Although we don't know if this prediction is correct because the authors don't actually measure the spread of depolarization experimentally (presumably because this is technically impossible), the model itself is a valuable advance. In particular, the model generated an interesting prediction – that GGN can only show the experimentally measured sustained odor-evoked depolarization if some KCs fire at high rates due to more intense/prolonged excitatory input from PNs – and the authors test the prediction experimentally, finding that a small previously overlooked fraction of KCs do in fact fire at high rates. Finally, the authors analyze the timing of IPSPs in GGN to infer that the "inhibitor of GGN" must be directly excited by odors (via KCs or PNs). While this KC->IG or PN->IG excitation was hypothesized in the original description of GGN (Papadopoulou et al), the timing of IPSPs in GGN provides novel evidence for this connectivity.

1) I am less convinced by Figure 3. This is certainly an interesting prediction of the model. But unlike Figures 4-6, which culminate in experimental verification of the prediction, here there is no experimental evidence that KCs dynamic range really is improved by inhibition. It's a prediction based on a model without experimental validation. While the model itself is useful, I feel that Figure 3 might be "a bridge too far". Could they experimentally test this model ? They could repeat Figure 3C of Papadopoulou et al., 2011 with a twist: use field stimulation to activate many KCs, while injecting varying amounts of current into one KC. The field stimulation should expand the dynamic range of the single KC that they inject current into, i.e. its spike rate as a function of current injected should have a shallower slope. Unfortunately, this would not test one of the weird results of the model, where the feedback inhibition prevents spiking at 14-16 pA even though the KC is not spiking and therefore can't trigger feedback inhibition, because this experiment isn't really feedback inhibition as defined in Figure 3, because the KC being measured doesn't affect its own activity at all (it's only being affected by the mass of other KCs). But, this can still be a useful result, that inhibition onto a KC expands KCs' dynamic range, and they could change their model to match the experiment. (The difficulty of testing their model directly as currently presented is itself a reason to change the model.) Plus, an increased dynamic range would suggest that GGN acts as "input gain control" rather than "response gain control" (Figure 3, Olsen et al., 2010 Neuron).

2) I also don't understand what happens in Figure 3B in the KC with GGN feedback at 14-16 pA. Presumably it doesn't spike (compared to the KC without GGN) because it's getting inhibition. But how can GGN be activated if the KC isn't spiking? Doesn't the KC need to spike at least once? If the model GGN releases GABA spontaneously, they should show that spontaneous inhibition isn't enough for increasing the dynamic range and you need odor-evoked activity in GGN.

3) Figure 5-6: Can the authors quantitatively compare the experimentally measured distribution of KC spike rates (6e) with the distribution in the model (5h)? How well do they match? And if they force the KC spike rates in the model to match the experimentally measured spike rates, does it reproduce sustained GGN odor responses? This is important to show that their experimental data match the computational prediction.

4) How can the authors be sure that their two outlier KCs in Figure 6E weren't weird experimental flukes? (Where they even 2 cells? maybe 2 recordings from the same cell?)

5) Could the synchronization of KC spikes in Figure 5C be alleviated by introducing variability in timing of KC inputs? For example, in Figure 7, KC->IG synapses have a 200 ms delay. Could KC inhibitory inputs also be delayed in some KCs more than others (more so than caused by the delay of passive spread of activity within GGN), eg based on the shape of KC dendrites? At least, ruling out this possibility is important for the implicit logic "they only way to get sustained GGN responses is if some KCs spike at high rates".

6) Figure 5F,G, 7G – I'd like to see the KC population spiking in these models (as in 5c). Presumably in Figure 5G the KCs won't be quite as synchronized as in 5c; and in 7g the KC spikes should precede the period of IG activity by 200 ms.

7) Figure 2—figure supplement 2, final paragraph of subsection “Signal attenuation in GGN” – does each KC get inhibition from exactly one GGN segment? That seems unrealistic. Also can the authors put error bars on # of spikes per KC?

8) Final paragraph of subsection “Signal attenuation in GGN”: “data not shown” – this seems like an interesting conclusion – can the authors make a figure of this?

9) “Earlier computational models(Kee et al., 2015; Papadopoulou et al., 2011) did not reproduce sustained responses in GGN” – this seems unfair to Kee et al., whose Figure 2D looks quite similar to Animal 1 in this manuscript's Figure 4, or the simulation in Figure 7G.

10) Materials and methods: “the synaptic conductances were selected from a lognormal distribution with the mean adjusted to produce realistic KC activity” Can the authors elaborate on this? Was the KC spiking threshold set to a particular value? What is “realistic KC activity”? How did they set the variance of the lognormal distribution?

Reviewer #2:

This manuscript describes a model of the locust olfactory system that is accompanied by electrophysiological recordings from the giant GABAergic neuron and the Kenyon cells. The model is constructed to test the role of the GGN in shaping KC output. The authors construct a model and then go through a series of steps to validate the model. Model validation was accomplished through observational recordings of the GGN or the KCs during olfactory stimulation.

As written, the manuscript reads like an unfolding story, where the authors create the model, test it, and then find it does not reproduce what is observed in the olfactory system. At each step, they alter a parameter of the model (e.g. synaptic strength of the KCs) to identify the contribution of this parameter to the output of the modelled neurons. This is informative, but I also found this style difficult to read. This style also makes the manuscript feel like it is not hypothesis driven. There are portions of the manuscript where the paragraphs ramble; the manuscript would be strengthened if the authors restructure these paragraphs. For example: they present a paragraph about how they had to adjust the model to include heterogeneity in synaptic strength and PN activation (subsection “GGN responses suggest some KCs fire at high rates”). But the reader doesn't find this conclusion until the end of the paragraph, and few details are reported as data in the paper. The paragraph should start out with the conclusion or a hypothesis and then present supporting evidence. The way it is written, the rationale feels ad hoc.

One of the main conclusions they claim from the model is that they discovered that the spike rates of some of the KCs was relatively high. (Introduction). As far as I could tell within the manuscript, the authors only tested a model in which the KC spike rates were all active or a model in which 1/5 of the neurons were inactive to show that when some of the KCs are silent and the synapse strength is heterologous, some KCs fire more than others. We know that the pattern of KC responses is odor-specific. It would have been good to test the model with different patterns of KC input (and presumably different proportions of inactive KCs) – as if stimulating with different odors – to show whether this outcome depended on the proportion of inactive KCs. A test of different synaptic strengths across the KCs would also have made this portion of the manuscript more convincing.

I thought the portion of this manuscript was the data/modelling of the IG neuron and its impact on the GGN was interesting, but I'm concerned about their conclusion that they have evidence for another olfactory pathway in the brain. The authors are forced to assume that the data of Papadapoulou et al. is correct and that spikes in GGN directly correlate with spikes in IG because they could not record from the IG neuron. However, when they cut the antennal nerve, they find that the GGN continues to spike – as normal – ruling out input from PNs or KCs to the IG. They conclude that the spiking in the IG is intrinsic or that it is generated from another source. What other olfactory source could it be, if it wasn't from the antenna….? Confusingly, later in the manuscript, they conclude that their model predicted that spiking in the IG could be driven by the KCs (Discussion paragraph seven). The contradictions in rationale need to be addressed and clarified.

I thought the discussion could be strengthened if the authors took more space in the discussion to compare their data/model to the recent paper on *Drosophila* from Inada et al., 2017.

Reviewer #3:

This study explores the function of the Giant GABAergic Neuron (GGN) in the locust using a combination of neural recordings and computational models. The authors conclude that the GGN contributes to an increase in the dynamic range of Kenyon Cells (KCs; the principle cells of the MB) and suggest the existence of a direct olfactory input pathway onto IG, an inhibitory neuron that directly synapses onto GGN. They report some novel functional properties of GGN (e.g. suppressive responses to odor), provide additional support for some known properties of GGN, and formulate some concrete hypotheses from their model that can be tested experimentally. Overall, this manuscript offers some additional detail about the function of this neuron within its network and some new testable hypotheses. My main concerns are twofold. (1) some of their findings are presented as novel when in fact they have already been shown before. These results need to be presented more clearly within the context of the existing literature. (2) some of their conclusions are not well-supported by the results that they provide. I detail these issues below.

1) Signal attenuation in GGN: The author's statement "Together, these features made it unclear whether this giant neuron has the biophysical capacity to perform its suggested function of carrying effective signals passively from the α lobe to distant points in the calyx" appears to not consider previous studies of this neuron that clearly show this capacity experimentally (Papadopoulou et al., 2011, Figure 3C), whereby activation of KCs is sufficient to propagate through the GGN to suppress KC spiking. Their anatomical characterization and compartmental model of GGN is a useful resource, but their conceptual findings should be more thoroughly contextualized in the existing knowledge of this neuron. Similarly, the authors should be more clear about the alternative hypotheses of GGN signal propagation their model is intended to disambiguate.

2) Subsection "Feedback inhibition expands the dynamic range of KCs". Given that this conclusion is entirely based on their model, this assertion should be changed to reflect that it is a strong and testable hypothesis, rather than an explicit test of that hypothesis. Moreover, the authors should better justify their model that uses a single KC with 50,000 synapses, rather than a more realistic model with 50,000 KCs. This seems like it would drive especially strong and synchronized (i.e. non-physiological) activation of GGN. Alternatively, since they subsequently enhance their model to include a larger complement of KCs, it would be relatively straightforward to consider that model here as well.

I am also a bit confused by the model of KC-GGN feedback to illustrate the capacity for expanding the dynamic range of KCs. The threshold for evoking spikes is increased when the KC is reciprocally connected to the GGN. The authors call this "feedback inhibition." But if KCs only influence the GGN via spikes (rather than graded release), then the GGN is not being activated by the KC with injected current ranging from 13-16pA (Figure 3B right). The only explanation I can see is that spontaneous activity from GGN is inhibiting the KC. However, if this is the case, then it's not really "feedback" inhibition, but just inhibition. If the authors want to claim that feedback inhibition is responsible for expanding the dynamic range of KCs, then they should determine what aspect of GGN activity is actually originating from the KC and not spontaneous activity. Because they have a detailed computational model, this analysis should be straightforward. Without it, their conclusion that feedback inhibition expands the dynamic range of KCs is not well supported by their model.

3) The finding that some KCs fire at high rates (subsection “GGN responses suggest some KCs fire at high rates”) also does not fully consider previous results. For instance, Perez-Orive et al., 2002, show that some KC-odor pairs (e.g. their Figure 1B and 2A) can elicit relatively high rates, as high as 16spikes/sec. Thus their predictions and data are entirely consistent with previously reported data, and their text should be updated to reflect this point, so that readers are aware of the relevance of existing literature. Perhaps more interestingly, I think that the authors have an excellent opportunity to test the function of these high-rate responses in the KC-GGN circuit. With their existing model, they could easily omit those high-rate KCs and observe the effects on the GGN. Are these particular responses critical for GGN operation? Do they serve any particular function? This would be relatively easy to explore and could unveil an unappreciated importance of these neurons.

Subsection “Odor evoked spiking in IG can explain GGN hyperpolarization”: Additional intuition for why the simple model of reciprocal connectivity between IG and GGN needs to be provided. The authors state that "a broad range of IG properties" was tested, but since this analysis is not shown, the reader cannot evaluate the capabilities of IG in driving odor-evoked hyperpolarization in GGN.

Final paragraph of subsection “Odor evoked spiking in IG can explain GGN hyperpolarization”: They authors state they "completely" silenced PNs and KCs by cutting the antennal nerves. Since locusts also have maxillary palps which contain ORNs that innervate the antennal lobe, I do not understand how cutting the antennal nerve silences all PNs and KCs. In principle, excitation from the palp could still provide enough excitation in PNs and KCs to drive spontaneous IPSPs in GGN. In order to make this claim, the authors need to additionally cut the maxillary nerve.

[Editors’ note: the decision after re-review follows.]

Thank you for resubmitting your work entitled "Feedback inhibition and its control in an insect olfactory circuit" for further consideration by *eLife*. Your revised article has been evaluated by Ronald Calabrese (Senior Editor), Kristin Scott (Reviewing Editor) and the three original reviewers.

The manuscript has been improved but there are some remaining issues that need to be addressed before acceptance, as outlined below:

1) Signal attenuation in GGN. The authors' justification in their response for why it is uncertain how signal may be propagating between KCs and GGN (based on prior data from Papadopoulou et al., for instance) makes sense to me. However, the text in the manuscript still is a bit sparse. I would suggest incorporating some of this explanation into the manuscript so that readers will understand the nuances of interpreting the results of the authors' model within the context of existing experimental data.

2) Experimental test of dynamic range expansion (Figure 3E-G). It is very difficult to assess the experimental data underlying the newly added results to test their theoretical finding that GGN expands the dynamic range of KCs. This is largely due to the fact that the experimental data are presented in such a highly processed form (Figure 3G) that it is impossible to assess directly. Ideally, more than two current levels would be injected into the KC (something more akin to the simulation in 3C). At the very least, the authors should show an example of the KC voltage traces (already measured) that went into Figure 3G in the main Figure 3 (not supplement) and make the raw data accessible as a Supplementary file. Given the details of the predictions in 3B, it is essential to show whether the texture of the experimental data reflects that of their model, to enable the reader to see similarities and any differences. Simply showing some of the voltage traces that they already have would help tremendously.

3) How did the authors determine the "high" and "low" amounts of current injected into GGN – were those amounts directly proportional to the "high" and "low" amounts of current injected into the KC?

4) Figure 3E, the color scheme could be clearer – perhaps the "A" pulses should be teal and the "B" pulses orange? And in 3G it would be clearer not to re-use colors that were used to indicate different models (eg purple and orange)

5) In Figure 3—figure supplement 1 I suggest using the same y-axis scale for 14 pA injection as for the other rows.

6) Discussion: "Given stronger odor stimuli (or in the absence of local KC-GGN reciprocal connections in the calyx), signals originating in the α lobe would produce inhibitory output throughout GGN's arbor, dominate local inhibition, and allow KCs receiving inhibition from the same region of GGN to fire together, reducing the contrast among their responses."

I had a hard time understanding the last part of this sentence. Why would global inhibition from GGN make KCs fire together – is it because KCs could only fire in time windows when GGN activity is low? If this only happens for KCs receiving inhibition from the same region of GGN, how is this different from local inhibition? And by "reducing the contrast among their responses" do they mean that global GGN inhibition is actually bad (for separating KC odor responses)? Perhaps the authors could clarify what they mean here.

[Editors' note: further revisions were suggested prior to acceptance, as described below.]

Thank you for resubmitting your work entitled "Feedback inhibition and its control in an insect olfactory circuit" for further consideration by *eLife*. Your revised article has been evaluated by Ronald Calabrese (Senior Editor), Kristin Scott (Reviewing Editor) and two of the original reviewers.

The manuscript has been improved but there are some remaining issues that need to be addressed before acceptance. As experimental evidence for dynamic range expansion is no longer included, the manuscript must be revised to discuss the model's predictions and limitations, as outlined below:

1) The authors have not demonstrated that "feedback inhibition expands the dynamic range of KCs" in the locust MB. Their conclusion about the dynamic range implies that they have experimental evidence, even though they are no longer including it. They need to make clear that their model predicts the dynamic range result, rather that stating definitively that feedback increases dynamic range.

2) As the authors now conclude with the model, I am concerned that they have provided neither a discussion of the potential caveats to interpretation of the model's predictions, nor any analysis of the robustness of their model's predictions to variations in parameters. An expanded discussion of the model's predictions and limitations would be valuable.

3) Abstract: "Our in vivo recordings and simulations of our model suggest that depolarization of GGN at its input branch can globally inhibit KCs several hundred microns away." This finding comes entirely from the model, not recordings.

4) Discussion paragraph five: they should change "revealed" to "predicts".

---

## [Author Response]

[Editors’ note: the authors submitted for reconsideration following the decision after peer review. What follows is the decision letter after the first round of review.]

Reviewer #1:Overall I like this combination of modelling and experiments – it covers an important topic and reaches interesting conclusions. I have some concerns that the authors should address before publication.Ray et al. construct a morphological model of the locust's giant GABAergic neuron (GGN) based on its reconstructed anatomy, and use it to model the spread of depolarization within GGN. The model predicts that although GGN is very large and doesn't spike, signals should still spread readily from one part to the other, though they get attenuated. Although we don't know if this prediction is correct because the authors don't actually measure the spread of depolarization experimentally (presumably because this is technically impossible), the model itself is a valuable advance. In particular, the model generated an interesting prediction – that GGN can only show the experimentally measured sustained odor-evoked depolarization if some KCs fire at high rates due to more intense/prolonged excitatory input from PNs – and the authors test the prediction experimentally, finding that a small previously overlooked fraction of KCs do in fact fire at high rates. Finally, the authors analyze the timing of IPSPs in GGN to infer that the "inhibitor of GGN" must be directly excited by odors (via KCs or PNs). While this KC->IG or PN->IG excitation was hypothesized in the original description of GGN (Papadopoulou et al), the timing of IPSPs in GGN provides novel evidence for this connectivity.

We thank the reviewer for finding our model “a valuable advance,” and for noting that it generated “an interesting prediction.” The reviewer is correct that testing the spread of depolarization within GGN in vivo proved technically impossible – several approaches we tried gave unclear results. Thus, we decided to use a computational model. We have now performed additional computational experiments, detailed below. Our new simulations extend our earlier conclusions. As we explain below, we also demonstrate high firing rates in KCs are an emergent property of the olfactory network, and contribute modestly to sustain odor-elicited activity in GGN.

1) I am less convinced by Figure 3. This is certainly an interesting prediction of the model. But unlike Figures 4-6, which culminate in experimental verification of the prediction, here there is no experimental evidence that KCs dynamic range really is improved by inhibition. It's a prediction based on a model without experimental validation. While the model itself is useful, I feel that Figure 3 might be "a bridge too far". Could they experimentally test this model ? They could repeat Figure 3C of Papadopoulou et al., 2011 with a twist: use field stimulation to activate many KCs, while injecting varying amounts of current into one KC. The field stimulation should expand the dynamic range of the single KC that they inject current into, i.e. its spike rate as a function of current injected should have a shallower slope. Unfortunately, this would not test one of the weird results of the model, where the feedback inhibition prevents spiking at 14-16 pA even though the KC is not spiking and therefore can't trigger feedback inhibition, because this experiment isn't really feedback inhibition as defined in Figure 3, because the KC being measured doesn't affect its own activity at all (it's only being affected by the mass of other KCs). But, this can still be a useful result, that inhibition onto a KC expands KCs' dynamic range, and they could change their model to match the experiment. (The difficulty of testing their model directly as currently presented is itself a reason to change the model.)

Following the reviewer’s suggestion, we have now performed additional experiments to test our prediction in vivo. As we describe below, the results confirm the prediction made from our model: increased inhibition originating in GGN onto a KC increases the dynamic range of the KC.

We tested the dynamic range of KCs by making intracellular recordings from them in vivo while injecting low or high amplitude current pulses. Simultaneously, we varied GGN-dependent inhibition onto the KCs by depolarizing GGN with intracellular current injections, also of low or high amplitude. Then, we measured the slope of KC firing rate in response to intracellular current stimuli in the KC with and without increased inhibition from GGN. (Details of this procedure are given in the text Materials and methods.)

We counted the number of spikes in each tested KC to obtain its average firing rate (f) in response to low and high current injections over multiple trials, with and without GGN depolarization. Then we computed the slope with respect to current amplitude by dividing the change in firing rate by the difference between the magnitudes of the two current levels (i1 and i2), i.e., [f(1,A) – f(2,A)] / (i1 – i2) for KC-only and [f(1,B) – f(2,B)]/(i1 – i2) for KC+GGN stimulation. Thus, a decrease in the slope upon GGN depolarization indicates an increase in the dynamic range of the KC.

In 5 of 5 experiments (p<.033, binomial test) we observed a decrease in the slope with GGN depolarization, consistent with the prediction from our model (shown in Figures 3C and D). We now provide this new information in Figure 3 E, F and G, and in the updated corresponding Results and Materials and methods sections.

As we now note in the text, our results are consistent with a general principle of Control Theory that negative feedback increases the linear range of a system [see, e.g. Astrom and Murray, “Feedback Systems”, v3.0i, Chapter 2, eqns 2.2 and 2.5. URL: http://www.cds.caltech.edu/~murray/books/AM08/pdf/fbs-principles_30Sep18.pdf]. Although our model is relatively complex with non-linear conductances generating spiking activity in KCs and graded negative feedback from GGN, this general principle seems to extend to this circuit. As can be seen in Figure 3C in our manuscript (and also in Figure 2.2 in Astrom and Murray) a wider dynamic range is accompanied by shallower slope in a plot of response vs input. In our case, the input is the current injected into a KC and the response is its spike rate.

Plus, an increased dynamic range would suggest that GGN acts as "input gain control" rather than "response gain control" (Figure 3, Olsen et al., 2010 Neuron).

We do not feel comfortable making a strong statement of input vs response gain control. The idea of “input gain control” and “response gain control” originated in visual neuroscience (McAdams and Maunsell, 1999; Reynolds and Desimone, 1999; Ayaz and Chance, 2009). The mathematical idea behind this differentiation seems to be to contrast input-output curves of the form

(1) y=a*x/(1+x) … (response gain control)

to those of the form

(2) y=x/(1+a+x) … (input gain control)

where x is input, and y is output or response. Curve (1) reaches the asymptotic value `a` for large `x`, thus the saturation level firing rate will be lower with inhibition (a < 1). Curve (2) reaches the asymptotic value of 1, same as without the factor `a`. In that sense our simulations suggest that GGN indeed modulates input gain, as the KC firing rates reach the same levels at extreme current injection. However, in our in vivo recordings we limited the current injection below saturation level for fear of damaging the cell. It is easy to show mathematically that the slope of both curves (1) and (2) are less than the curve with unity gain: x/(1+x) for 0 < a < 1. Thus, a change in slope alone is insufficient for distinguishing between the two.

We discuss the absence of spiking in 14-16 pA in the next comment.

2) I also don't understand what happens in Figure 3B in the KC with GGN feedback at 14-16 pA. Presumably it doesn't spike (compared to the KC without GGN) because it's getting inhibition. But how can GGN be activated if the KC isn't spiking? Doesn't the KC need to spike at least once? If the model GGN releases GABA spontaneously, they should show that spontaneous inhibition isn't enough for increasing the dynamic range and you need odor-evoked activity in GGN.

The GGN model indeed includes some inhibition in the absence of odorant. This is meant to reflect conditions in vivo. As shown in Figure 9J, GGN’s membrane potential activity changes very little after the antenna is cut, a manipulation known to silence PNs and KCs (Joseph et al., 2012). This observation suggests GGN itself is spontaneously active, or that it receives input from other neurons that are themselves spontaneously active or activated by stimuli other than odors. To clarify our results we have now updated Figures 3B and c with model results showing the effect of spontaneous inhibition from GGN on a single KC when the KC-to-GGN connection is removed. Spontaneous inhibition increases the spiking threshold of the KC, and slightly increases its dynamic range (purple traces). By contrast, the complete feedback circuit increases the dynamic range of the KC much more (orange traces). To make it clear that GGN provides inhibition to KCs even at rest we updated the text as follows:

“As expected, baseline inhibition from spontaneous activity in GGN increased the KC’s threshold for spiking.”

For increased clarity we have also replotted Figure 3C.

3) Figure 5-6: Can the authors quantitatively compare the experimentally measured distribution of KC spike rates (6e) with the distribution in the model (5h)? How well do they match? And if they force the KC spike rates in the model to match the experimentally measured spike rates, does it reproduce sustained GGN odor responses? This is important to show that their experimental data match the computational prediction.

We thank the reviewer for this interesting question that led us to test additional simulations.

Our initial simulations generated distributions of KC spike rates qualitatively similar to distributions we obtained in vivoin that both had long rightward tails. However, standard statistical tests (Mann-Whitney U and Kolmogorov-Smirnov) showed that the *in silico* and in vivo distributions were not identical. Further modeling in which we simulated many instances of the mushroom body network (using different random number generator seeds such that individual synaptic connections and strengths were randomized but, overall, these features kept the same mean and standard deviation across the networks) yielded KC firing rate distributions that were qualitatively similar to each other, but, as above, statistical tests showed that these distributions were not identical. This result shows that the exact distribution of KC firing rates is a complex emergent property of each instantiation of the network. The result also shows it would be difficult or impossible to force the KC spike rates in our model to exactly match the experimentally measured spike rates. To make this state of affairs clear to readers we have revised the text and have added a new supplementary figure, Figure 6—figure supplement 1, to illustrate more examples of KC spike count distributions and their associated GGN membrane voltages.

4) How can the authors be sure that their two outlier KCs in Figure 6E weren't weird experimental flukes? (Where they even 2 cells? maybe 2 recordings from the same cell?)

The two outliers in Figure 6E are indeed two different Kenyon cells; they were separately visualized and targeted with patch electrodes, filled with dye and reconstructed. We now emphasize this in the legend. We don’t think these results are weird experimental flukes because, as we should have made clear in our manuscript, others have obtained similar results: see Figure 2A in Perez-Orive et al., 2002 and Figures 1, B1 and B2 in Gupta and Stopfer, 2014. We now cite these previous observations in our manuscript: “Although relatively rare KC odor responses with many spikes have been observed before (Gupta and Stopfer, 2014; Perez-Orive et al., 2002), the origin and significance of these responses were unclear.”

5) Could the synchronization of KC spikes in Figure 5C be alleviated by introducing variability in timing of KC inputs? For example, in Figure 7, KC->IG synapses have a 200 ms delay. Could KC inhibitory inputs also be delayed in some KCs more than others (more so than caused by the delay of passive spread of activity within GGN), eg based on the shape of KC dendrites? At least, ruling out this possibility is important for the implicit logic "they only way to get sustained GGN responses is if some KCs spike at high rates".

We thank the reviewer for this interesting suggestion which led us to perform additional modeling experiments. As suggested, we investigated whether variability in synaptic delays to KCs could alleviate their overly-synchronized spiking. Instead of introducing additional delays in the graded synapses from GGN to KCs (which would be difficult to implement), we introduced a set of variable synaptic delays in the PN-to-KC connections and simulated the model with PN input patterns similar to those shown in Figure 5B. However, we found the variations we tested in this arrangement produced overly-synchronized spiking in KCs (see Figure 5—figure supplement 1) and did not generate sustained GGN responses. We want to clarify that having some KCs spike at high rates alone is not enough to generate realistic responses in GGN. For example, we also tried introducing some KCs firing at high rates by setting high PN→KC conductances in simulations that did not include temporal patterning of PN activity. These simulations, however, also yielded KCs whose firing was overly-synchronized, resulting in unrealistic oscillatory peaks in GGN’s membrane potential.

To clarify this we have updated the text in the Results as follows:

“Increasing the diversity of synaptic delays from PNs to KCs did not alleviate overly-synchronized spiking in the KC population if heterogeneity in synaptic strengths and temporal patterns of PN activity were omitted from the model (Figure 5—figure supplement 1).”

We updated the Materials and methods section as follows:

“To test whether diverse synaptic delays could alleviate synchronized spiking in KC population, we simulated a set of networks where PN -> KC synapses were assigned a range of delays from a normal distribution with mean 7.5 ms and standard deviation 3 ms. We chose these values with the animal’s physiology in mind, by assuming a conservative speed of 0.2 m/s = 200 μm/ms for signal propagation through dendrites (Larkum, Rioult, and Luscher, 1996). The length of PN axon is about 1500 µm, so the total time delay from PN to KCs should be under 1500 / 200 = 7.5 ms (also, (Jortner et al., 2007) found a delay of about 6ms). The maximal diameter of a large KC dendritic arbor is about 600µm in the calyx and thus, the longest delay between two KCs receiving spikes from the same PN in their soma should be about 3 ms.”

6) Figure 5F,G, 7G – I'd like to see the KC population spiking in these models (as in 5C). Presumably in Figure 5G the KCs won't be quite as synchronized as in 5C; and in 7G the KC spikes should precede the period of IG activity by 200 ms.

The reviewer is correct in both points: KCs are less synchronized, and KC spikes precede the period of IG activity by 200 ms. We have updated Figure 5 to show KC spike rasters and have moved the KC histogram to Figure 6. (We’ve also corrected a small error in Figure 5E-G: we originally gave off-response duration as 200 ms, now corrected to 500 ms).

Figure 7 is now Figure 8. We redesigned this figure by adding KC spike rasters and splitting off other content to a new Figure 9. Figure 8 now shows GGN’s membrane potential in vivo with IPSPs and the inferred PSTH of IG spikes. New Figure 9 shows the simulated GGN and IG membrane potentials along with PN and KC spike rasters, and the spontaneous IPSPs recorded in vivo from GGNs of animals with intact or cut antennal nerves.

7) Figure 2—figure supplement 2, final paragraph of subsection “Signal attenuation in GGN” – does each KC get inhibition from exactly one GGN segment? That seems unrealistic.

In our model, indeed, each KC receives inhibition from one GGN segment. The number and locations of GGN’s synapses onto individual KCs are unknown, but in the locust, KCs have multiple dendritic branches, so it is likely that each KC receives GGN input from multiple regions of the calyx. In our model we merged all possible GGN-KC synapses into a single synapse per KC as a manageable simplification.

This simplification does not compromise our result that the effect of voltage differences distributed across GGN’s arbor on KC activity is small. With a single synapse per KC, two KCs can receive quite different levels of inhibition if the presynaptic terminals are on GGN neurites carrying different voltages. On the other hand, if a KC receives inhibition at multiple dendritic branches contacting different parts of GGN’s arbor that carry different voltages, then the total effect observed at the spike initiation zone should be a superposition of the effects due to each synapse, reducing the variation between different KCs. As we point out in the next line in the manuscript, this is a small effect either way.

Also, after adding a figure showing the distribution of signal transmission delays across GGN.

Also can the authors put error bars on # of spikes per KC?

Agreed and done.

8) Final paragraph of subsection “Signal attenuation in GGN”: “data not shown” – this seems like an interesting conclusion – can the authors make a figure of this?

Yes, we have now added Figure 2—figure supplement 3 showing Spearman’s rank correlation coefficient for the different parameters influencing KC spiking in three different sets of simulations. (Earlier Supplementary Figure S4 is now Figure 5—figure supplement 2)

9) “Earlier computational models (Kee et al., 2015; Papadopoulou et al., 2011) did not reproduce sustained responses in GGN” – this seems unfair to Kee et al., whose Figure 2D looks quite similar to Animal 1 in this manuscript's Figure 4, or the simulation in Figure 7G.

We agree and have now rephrased this to more specifically describe our observations:

“Earlier computational models (Papadopoulou et al., 2011; Kee et al., 2015) did not attempt to reproduce the variety of GGN responses we observed in vivo, including hyperpolarizations.”

10) Materials and methods: “the synaptic conductances were selected from a lognormal distribution with the mean adjusted to produce realistic KC activity” Can the authors elaborate on this? Was the KC spiking threshold set to a particular value? What is “realistic KC activity”? How did they set the variance of the lognormal distribution?

These are good questions. We have now updated our Materials and methods section with more details. Briefly, in our model KC the spike generation mechanism is implemented as a set of Hodgkin-Huxley (HH) type ionic conductances. In contrast to simpler models such as Integrate and Fire, HH type models do not have an explicit threshold voltage; rather, the threshold emerges from channel dynamics (Izhikevich, 2007, Dynamical Systems in Neuroscience, p 238 ff). So, rather than setting an explicit threshold value we carried out a parameter search to obtain a preliminary estimate of PN→KC synaptic conductances sufficient to evoke a few spikes in a single KC model. Later, in the full network model, we tested values around this conductance along with a range of GGN→KC conductances. In the network simulations we considered KC activity to be realistic if around 10% or fewer neurons in the KC population spiked in response to odor stimulus, as has been documented in vivo. Also, we considered KC activity too low to evoke depolarization in GGN to be unrealistic. In the lognormal distributions for synaptic strengths, the standard deviation was set to (1.0 x mean) to follow roughly Song et al., 2005.

Reviewer #2:This manuscript describes a model of the locust olfactory system that is accompanied by electrophysiological recordings from the giant GABAergic neuron and the Kenyon cells. The model is constructed to test the role of the GGN in shaping KC output. The authors construct a model and then go through a series of steps to validate the model. Model validation was accomplished through observational recordings of the GGN or the KCs during olfactory stimulation.As written, the manuscript reads like an unfolding story, where the authors create the model, test it, and then find it does not reproduce what is observed in the olfactory system. At each step, they alter a parameter of the model (e.g. synaptic strength of the KCs) to identify the contribution of this parameter to the output of the modelled neurons. This is informative, but I also found this style difficult to read. This style also makes the manuscript feel like it is not hypothesis driven. There are portions of the manuscript where the paragraphs ramble; the manuscript would be strengthened if the authors restructure these paragraphs. For example: they present a paragraph about how they had to adjust the model to include heterogeneity in synaptic strength and PN activation (subsection “GGN responses suggest some KCs fire at high rates”). But the reader doesn't find this conclusion until the end of the paragraph, and few details are reported as data in the paper. The paragraph should start out with the conclusion or a hypothesis and then present supporting evidence. The way it is written, the rationale feels ad hoc.

We appreciate these suggestions and have revised our text throughout for clarity. Our goals in organizing our manuscript have been to be informative and to make our reasoning as clear as possible. To address the specific example provided by the reviewer, we have changed the title of the first subsection in Results section to “Signals travelling through GGN’s arbor attenuate significantly,” and have updated this section to state our initial hypothesis at the beginning. The rest of the subsections are now titled with their respective conclusions.

One of the main conclusions they claim from the model is that they discovered that the spike rates of some of the KCs was relatively high. (Introduction). As far as I could tell within the manuscript, the authors only tested a model in which the KC spike rates were all active or a model in which 1/5 of the neurons were inactive to show that when some of the KCs are silent and the synapse strength is heterologous, some KCs fire more than others. We know that the pattern of KC responses is odor-specific. It would have been good to test the model with different patterns of KC input (and presumably different proportions of inactive KCs) – as if stimulating with different odors – to show whether this outcome depended on the proportion of inactive KCs. A test of different synaptic strengths across the KCs would also have made this portion of the manuscript more convincing.

Here we think the reviewer may have mistaken results from PNs for results from KCs. We have now revised our text to make this clear.

As we show in Figures 5 (B) and (E), we activated either 1/5^th^ or 2/3^rd^ of PNs, not KCs, to simulate the odor-elicited input ultimately reaching KCs. Although the compressed nature of the figure may give the impression that KC activity in Figures 5 (C), (F) and (H) was dense, our model was constrained such that only about 5000, or 10% of the 50000 KCs, were spiking.

We did indeed simulate different odor responses by changing the temporal patterns of PN activity in the model (see Figure 9), but we realize this was not clear enough from the text and the figures. We have now added spike rasters of the PN population for two different temporal patterns in Figure 9 (B) and (F).

I thought the portion of this manuscript was the data/modelling of the IG neuron and its impact on the GGN was interesting, but I'm concerned about their conclusion that they have evidence for another olfactory pathway in the brain. The authors are forced to assume that the data of Papadapoulou et al. is correct and that spikes in GGN directly correlate with spikes in IG because they could not record from the IG neuron. However, when they cut the antennal nerve, they find that the GGN continues to spike – as normal – ruling out input from PNs or KCs to the IG. They conclude that the spiking in the IG is intrinsic or that it is generated from another source. What other olfactory source could it be, if it wasn't from the antenna….? Confusingly, later in the manuscript, they conclude that their model predicted that spiking in the IG could be driven by the KCs (Discussion paragraph seven). The contradictions in rationale need to be addressed and clarified.

Again, we have revised our text to clarify a point of confusion. The possibility of spontaneous activity in IG does not contradict the hypothesis that KCs excite IG. A cell can be spontaneously active (see Hausser et al., 2004. J. Neurosci.), and this activity can increase following stimulation; for example, ORNs are spontaneously active, but odor presentations increase (or decrease) this activity (Joseph et al., 2012). The drive for spontaneous activity in IG may be intrinsic or it may arise in sensory modalities other than olfaction, such as vision. Here is our line of thought: Because KCs require strong and coincident input from PNs to spike, spontaneous activity in the antennal lobe evokes little activity in KCs (Laurent and Naraghi, 1994; Perez-Orive et al., 2002; Joseph et al., 2012) and thus does not affect spontaneous activity in IG. Odor stimulation, however, increases activity in the antennal lobe, raising the KCs above spiking threshold and generating activity across the KC population, which, in turn, drives an increase in IG spiking.

Another possibility, as pointed out by reviewer # 1 in comment # 13, is that even extremely rare spontaneous activity in KCs, when summed over a population of 50,000 cells, might drive spontaneous spiking in IG. However, as we explain above, we believe KCs are completely silent after antennal nerves are cut (Joseph et al., 2012), but we are unable to test this directly.

We have now updated our text to make this clear: “This result leads us to hypothesize that KCs, which require strong and coincident input from PNs to spike (Perez-Orive et al., 2002; Jortner et al., 2007) may serve as thresholding elements in the odor pathway from the antennal lobe to IG, transferring odor-evoked activity, but not lower levels of spontaneous activity, from PNs.”

And “KCs are nearly silent at rest, making them unlikely direct sources of this activity, but it is possible that even extremely sparse spontaneous activity distributed over the population of 50,000 KCs could suffice to drive spontaneous spiking in IG. This possibility would be difficult to test in vivo.”

I thought the discussion could be strengthened if the authors took more space in the discussion to compare their data/model to the recent paper on *Drosophila* from Inada et al., 2017.

We agree and have updated this part with a more detailed discussion of Inada et al., 2017 and how our results relate to their findings.

Reviewer #3:[…] My main concerns are twofold. (1) some of their findings are presented as novel when in fact they have already been shown before. These results need to be presented more clearly within the context of the existing literature. (2) some of their conclusions are not well-supported by the results that they provide. I detail these issues below.1) Signal attenuation in GGN: The author's statement "Together, these features made it unclear whether this giant neuron has the biophysical capacity to perform its suggested function of carrying effective signals passively from the α lobe to distant points in the calyx" appears to not consider previous studies of this neuron that clearly show this capacity experimentally (Papadopoulou et al. 2011, Figure 3C), whereby activation of KCs is sufficient to propagate through the GGN to suppress KC spiking. Their anatomical characterization and compartmental model of GGN is a useful resource, but their conceptual findings should be more thoroughly contextualized in the existing knowledge of this neuron. Similarly, the authors should be more clear about the alternative hypotheses of GGN signal propagation their model is intended to disambiguate.

Our work considers and fully incorporates earlier findings that GGN effectively inhibits KCs through a feedback mechanism. Papadopoulou et al., 2011 suggested, based on the morphologies of GGN’s neurites, that the neuron receives input from KCs in its α lobe branches and sends its output via its calyceal branches. However, nothing in Papadopoulou et al. excludes the possibility that feedback from GGN to KCs occurs through local interactions within the calyx. As we explain in our text, given the non-spiking nature of GGN, the distance between the α lobe and the calyx, and other factors, we hypothesized that depolarizing signals originating in the α lobe will greatly attenuate by the time they reach the calyx, possibly to the point of being ineffective. If this idea is correct, then the inhibition of KCs by GGN demonstrated by Papadopoulou et al. would have to occur via a different mechanism, such as through reciprocal connections between KCs and GGN within the calyx. Several other lines of evidence are consistent with our idea. Papadopoulou et al. noted that activating GGN synaptically was more effective than activating it through direct current injection in counteracting current-induced spiking of KC. Assuming Papadopoulou et al. impaled GGN in one of its relatively thick branches in the peduncle or the α lobe (these are by far the most successful target regions), their observation is consistent with the idea that effective local feedback may occur within calyx. Further, a recent study found evidence for pre- and post-synaptic terminals within calyceal branches of the APL neuron, the *Drosophila*homologue of GGN (Zheng et al., 2018).

Thus, the goal of our simulations was to investigate whether the extent of signal attenuation in GGN would prioritize the idea of local rather than global feedback, a possibility that would raise interesting questions about the spatial structure or lack thereof in KC↔GGN connections. Our model suggested that, although signals originating in GGN’s α lobe branches do attenuate significantly, enough signal remains in the calyx to hyperpolarize KCs there. This finding made further exploration of alternative hypotheses of GGN signal propagation moot.

2) Subsection "Feedback inhibition expands the dynamic range of KCs". Given that this conclusion is entirely based on their model, this assertion should be changed to reflect that it is a strong and testable hypothesis, rather than an explicit test of that hypothesis.

Yes, we agree. We have now tested this prediction in vivo, and the results support our hypothesis. The results are described above in a response to reviewer 1 and in the revised manuscript

Moreover, the authors should better justify their model that uses a single KC with 50,000 synapses, rather than a more realistic model with 50,000 KCs. This seems like it would drive especially strong and synchronized (i.e. non-physiological) activation of GGN. Alternatively, since they subsequently enhance their model to include a larger complement of KCs, it would be relatively straightforward to consider that model here as well.

The goal of our simulations with a single KC was to isolate the specific effect of negative feedback from the complexity generated by the full network. As we describe in the text, we avoided synchronous input by jittering the synaptic delays. To clarify this we have now revised the manuscript as follows:

“Moreover, to avoid unrealistic, strong synchronous input to GGN we jittered the incoming spike times by applying random synaptic delays between 0 and 60ms. Thus, after each spike generated by the model KC, GGN received 50,000 EPSPs distributed over a 60ms time window.”

In addition, we took the reviewer’s suggestion and carried out new simulations. In the full-scale network, the spiking of a single KC would have negligible effect on GGN depolarization. Thus, testing a single KC receiving inhibition with the same time course as in an extended network simulation should closely approximate its response in the full-scale model. Therefore, we simulated in our model a dynamic clamp experiment: we played the GGN membrane potential from a simulation of the extended network to a single KC via a graded inhibitory synapse. We injected a range of currents to this KC in the same way as shown in Figure 3. The results were consistent with our above findings, as shown in a new supplementary figure, Figure 3—figure supplement 1.

I am also a bit confused by the model of KC-GGN feedback to illustrate the capacity for expanding the dynamic range of KCs. The threshold for evoking spikes is increased when the KC is reciprocally connected to the GGN. The authors call this "feedback inhibition." But if KCs only influence the GGN via spikes (rather than graded release), then the GGN is not being activated by the KC with injected current ranging from 13-16pA (Figure 3B right). The only explanation I can see is that spontaneous activity from GGN is inhibiting the KC. However, if this is the case, then it's not really "feedback" inhibition, but just inhibition. If the authors want to claim that feedback inhibition is responsible for expanding the dynamic range of KCs, then they should determine what aspect of GGN activity is actually originating from the KC and not spontaneous activity. Because they have a detailed computational model, this analysis should be straightforward. Without it, their conclusion that feedback inhibition expands the dynamic range of KCs is not well supported by their model.

Reviewer 1 raised this point as well (see comment # 2). As we describe in detail above, reflecting the situation in vivo, the model does indeed include spontaneous inhibition, but this inhibition is greatly enhanced by feedback from KC activity. To clarify this point, we have now run additional simulations, revised our text, and added new figures (Figure 3B, middle panel and updated Figure 3C) showing responses of KCs when they receive only spontaneous inhibition from GGN, with no feedback inhibition. These results provide strong support for our conclusion that feedback inhibition expands the dynamic range of KCs.

3) The finding that some KCs fire at high rates (subsection “GGN responses suggest some KCs fire at high rates”) also does not fully consider previous results. For instance, Perez-Orive et al., 2002 Science, show that some KC-odor pairs (e.g. their Figure 1B and 2A) can elicit relatively high rates, as high as 16spikes/sec. Thus their predictions and data are entirely consistent with previously reported data, and their text should be updated to reflect this point, so that readers are aware of the relevance of existing literature. Perhaps more interestingly, I think that the authors have an excellent opportunity to test the function of these high-rate responses in the KC-GGN circuit. With their existing model, they could easily omit those high-rate KCs and observe the effects on the GGN. Are these particular responses critical for GGN operation? Do they serve any particular function? This would be relatively easy to explore and could unveil an unappreciated importance of these neurons.

We thank the reviewer (and reviewer 1 as well) for highlighting these previous results. We have now revised our text to discuss them.

We have also run new simulations, as suggested by the reviewer, to explore questions about high-firing-rate KCs. We found that these KCs inevitably emerge from activity in the feedback network, and that the identities of high-firing-rate KCs change from odor to odor in any given network. In the simplest model specification, we only set the mean and standard deviation of the distribution from which the synaptic strengths were sampled. Thus, we could reduce the mean strength of PN→KC synapses and/or increase the same for GGN→KC synapses to restrict the entire KC population to fire only small numbers of spikes. In this condition we found the KCs began to spike in extreme, unrealistic synchrony, and failed to produce sustained depolarization of GGN.

We further followed the suggestion of the reviewer and simulated 10 network model instances including the one used in Figure 5I, but after specifically removing the excitatory synapses to all the KCs that had produced more than 5 spikes in the original simulation, thus effectively deleting them from the network. Notably, removing these high-firing-rate KCs did not produce substantial differences in GGN’s response because another set of KCs arose within the network to replace them and became high-spiking. However, the number of high spiking KCs in the modified model was smaller than the original. Therefore, we ran a reiterative series of simulations, removing the KCs that produced more than 5 spikes in the previous model until no remaining KCs fulfilled the removal criterion. As high-spike rate KCs were removed from the network, the simulated GGN response changed from realistic levels of sustained depolarization to widely-separated peaks of activity, suggesting these KCs augment GGN’s depolarizing response. We added these results as new Figure 7. To be clear, we did not explicitly specify these firing rates into our model KCs – rather, as appears to be the case in vivo, they emerged on their own. This convergence of observations made in vivo and, in our model, gives us confidence that our model is reasonably realistic. We have revised our text to describe these new analyses, and have updated the Discussion section to clarify the emergence and roles of high-spike rate KCs.

Subsection “Odor evoked spiking in IG can explain GGN hyperpolarization”: Additional intuition for why the simple model of reciprocal connectivity between IG and GGN needs to be provided. The authors state that "a broad range of IG properties" was tested, but since this analysis is not shown, the reader cannot evaluate the capabilities of IG in driving odor-evoked hyperpolarization in GGN.

We agree and have now updated this section with the following more intuitive explanation of our reasoning:

“We tested a range of synaptic strengths and time constants for GGN→IG, and IG→GGN synapses, and manipulated baseline firing rates of IG, but none of these efforts could generate odor-elicited hyperpolarization in GGN (data not shown). This hinted at a discrepancy in the causal connection between GGN and IG activity. Intuitively, if IG spiking increased solely because inhibition from GGN decreased, then IG spiking could only increase after GGN returned to baseline. However, as evident in Figure 8, sometimes GGN, recorded in vivo, hyperpolarized at the odor’s onset, ruling out the possibility that disinhibition from GGN is the sole driver of spiking in IG.”

Final paragraph of subsection “Odor evoked spiking in IG can explain GGN hyperpolarization”: They authors state they "completely" silenced PNs and KCs by cutting the antennal nerves. Since locusts also have maxillary palps which contain ORNs that innervate the antennal lobe, I do not understand how cutting the antennal nerve silences all PNs and KCs. In principle, excitation from the palp could still provide enough excitation in PNs and KCs to drive spontaneous IPSPs in GGN. In order to make this claim, the authors need to additionally cut the maxillary nerve.

The reviewer is correct, but facts we inadvertently omitted convince us that palp input did not contribute to our results. The locust preparation we used here included cutting the labral nerves and removing the suboesophageal ganglion. In previous work (Joseph et al., 2012) we confirmed that cutting the antenna in this preparation does in fact completely silence PNs and KCs. We have now clarified this in the Materials and methods:

“Our dissection included cutting the labral nerves and removing the suboesophageal ganglion (SOG), precluding stimulation via olfactory receptors in the palps.”

We also made this explicit in the text, now reads:

“… we completely silenced PNs and KCs by bilaterally cutting the animal’s antennal and labral nerves (Joseph et al., 2012) …”

[Editors’ note: the decision after re-review follows.]

The manuscript has been improved but there are some remaining issues that need to be addressed before acceptance, as outlined below:1) Signal attenuation in GGN. The authors' justification in their response for why it is uncertain how signal may be propagating between KCs and GGN (based on prior data from Papadopoulou et al., for instance) makes sense to me. However, the text in the manuscript still is a bit sparse. I would suggest incorporating some of this explanation into the manuscript so that readers will understand the nuances of interpreting the results of the authors' model within the context of existing experimental data.

We agree and have now brought more of this explanation into the Introduction:

“If signals originating in GGN’s α lobe attenuate to the extent that they cannot effectively hyperpolarize KCs, then GGN must be exerting its influence on KCs through a different mechanism. One such possibility is that GGN could inhibit KCs through local interactions entirely within the mushroom body calyx. Consistent with this idea, synaptically stimulating GGN has been found to be more effective than direct depolarization by intracellular current injection in counteracting current induced spiking of a KC (Papadopoulouet al., 2011). Also, APL, the *Drosophila*homologue of GGN, has been found to have both pre- and post-synaptic terminals within its calyceal branches(Zhenget al., 2018). These findings lend support to the hypothesis that GGN may inhibit KCs via local reciprocal connections within the calyx.”

2) Experimental test of dynamic range expansion (Figure 3E-G). It is very difficult to assess the experimental data underlying the newly added results to test their theoretical finding that GGN expands the dynamic range of KCs. This is largely due to the fact that the experimental data are presented in such a highly processed form (Figure 3G) that it is impossible to assess directly. Ideally, more than two current levels would be injected into the KC (something more akin to the simulation in 3c). At the very least, the authors should show an example of the KC voltage traces (already measured) that went into Figure 3G in the main Figure 3 (not supplement) and make the raw data accessible as a Supplementary file. Given the details of the predictions in 3b, it is essential to show whether the texture of the experimental data reflects that of their model, to enable the reader to see similarities and any differences. Simply showing some of the voltage traces that they already have would help tremendously.

Since the last submission of our manuscript, hoping to increase our “N” and obtain a compelling illustration, we performed additional replicates of the in vivo experiment to test KC dynamic range. Unfortunately, the new results have convinced us that we lack the experimental control needed to appropriately test this prediction. In brief, the experiments described by the reviewer require prolonged paired recordings and a long series of intracellular current injections in tiny Kenyon cells and, simultaneously, in the unique neuron GGN. We have learned that, in the context of this experiment, small variations in the membrane potentials of either neuron lead to uninterpretable results. Because we cannot perform the necessary tests, we now conclude this part of the manuscript with a statement of our model’s prediction. We believe this is appropriate for several reasons: it is not unusual for authors of a computational model to conclude their work with predictions; as we point out (here and in the manuscript) our model’s prediction is consistent with basic principles of control theory; and comparable experimental results have been reported in other model systems where more stable preparations were possible (Olsen et al., 2010 Neuron; Inada et al., 2017). We note that this prediction of our model represents only a small part of our manuscript.

3) How did the authors determine the "high" and "low" amounts of current injected into GGN – were those amounts directly proportional to the "high" and "low" amounts of current injected into the KC?

We have now removed this experimental test from our manuscript. Please see our response to comment #2, above.

4) Figure 3E, the color scheme could be clearer – perhaps the "A" pulses should be teal and the "B" pulses orange? And in 3g it would be clearer not to re-use colors that were used to indicate different models (eg purple and orange)

Please see response to comment #2 above. We have now removed this experimental test and corresponding Figures 3E, F and G from our manuscript.

5) In Figure 3—figure supplement 2 I suggest using the same y-axis scale for 14 pA injection as for the other rows.

We agree and have updated the figure as suggested.

6) Discussion: "Given stronger odor stimuli (or in the absence of local KC-GGN reciprocal connections in the calyx), signals originating in the α lobe would produce inhibitory output throughout GGN's arbor, dominate local inhibition, and allow KCs receiving inhibition from the same region of GGN to fire together, reducing the contrast among their responses."I had a hard time understanding the last part of this sentence. Why would global inhibition from GGN make KCs fire together – is it because KCs could only fire in time windows when GGN activity is low? If this only happens for KCs receiving inhibition from the same region of GGN, how is this different from local inhibition? And by "reducing the contrast among their responses" do they mean that global GGN inhibition is actually bad (for separating KC odor responses)? Perhaps the authors could clarify what they mean here.

We thank the reviewers for helping us clarify our meaning. Our intention was to note that local inhibition could enhance the influence of weak input, not to argue that global inhibition would somehow be bad. We have now rephrased this as follows:

“Assuming randomly-connected input from PNs to KCs(Jortneret al., 2007; Caronet al., 2013; Eichleret al., 2017), we can speculate about possible impact of strong or weak odor stimuli on the circuit. Strong odor stimuli might drive this circuitry into a global, population-wide, winner-take-all scenario in which only KCs receiving the strongest inputs manage to spike. Weak stimuli, on the other hand, might recruit only local inhibition, allowing a distribution of winners to emerge locally, resulting in a center-surround-type of contrast enhancement.”

[Editors' note: further revisions were suggested prior to acceptance, as described below.]

The manuscript has been improved but there are some remaining issues that need to be addressed before acceptance. As experimental evidence for dynamic range expansion is no longer included, the manuscript must be revised to discuss the model's predictions and limitations, as outlined below:1) The authors have not demonstrated that "feedback inhibition expands the dynamic range of KCs" in the locust MB. Their conclusion about the dynamic range implies that they have experimental evidence, even though they are no longer including it. They need to make clear that their model predicts the dynamic range result, rather that stating definitively that feedback increases dynamic range.

We agree. We want to be perfectly clear that we are offering a prediction about dynamic range expansion and have now revised our manuscript to ensure this. Specifically, we changed the title of this Results section to: “Feedback inhibition is predicted to expand the dynamic range of KCs;” and, as noted below, in the Discussion, we changed “revealed” to “predicts.”

2) As the authors now conclude with the model, I am concerned that they have provided neither a discussion of the potential caveats to interpretation of the model's predictions, nor any analysis of the robustness of their model's predictions to variations in parameters. An expanded discussion of the model's predictions and limitations would be valuable.

We agree and have now expanded the Discussion with the following paragraph about caveats to our model and its predictions:

“Several caveats apply to predictions of our model. Although we explored a wide range of values for GGN’s membrane and cytoplasmic conductances to test our model for robustness (Figure 2G), we cannot exclude the possibility that ion channels are distributed unevenly across GGN, potentially allowing parts of the neuron to behave in ways we could not observe in our recordings. We simplified our models of KCs as single compartments, and in some cases collapsed multiple synapses into a single equivalent synapse; however, these simplifications should not affect our predictions in a qualitative sense. Finally, our model excludes plasticity mechanisms, so our predictions apply best to odors that are novel or delivered at long intervals.”

3) Abstract: "Our in vivo recordings and simulations of our model suggest that depolarization of GGN at its input branch can globally inhibit KCs several hundred microns away." This finding comes entirely from the model, not recordings.

We agree and have updated the Abstract to correct this statement:

“Our simulations suggest that depolarization of GGN at its input branch can globally inhibit KCs several hundred microns away. Our in vivo recordings show that GGN responds to odors with complex temporal patterns of depolarization and hyperpolarization that can vary with odors and across animals, leading our model to predict the existence of a yet-undiscovered olfactory pathway.”

4) Discussion paragraph five: they should change "revealed" to "predicts".

We agree and have updated this sentence accordingly:

“Notably, our model also predicts that feedback inhibition from GGN can expand the dynamic range of inputs able to activate KCs (Figure 3).”